# Circular RNAs in the human brain are tailored to neuron identity and neuropsychiatric disease

Xianjun Dong [1,2,3,4], Yunfei Bai [1,2,5], Zhixiang Liao[1,2], David Gritsch[1,2], Xiaoli Liu[1,2,6], Tao Wang [1,2,7], Rebeca Borges-Monroy [1,2], Alyssa Ehrlich[1,2,8], Geidy E. Serrano[9], Mel B. Feany [10], Thomas G. Beach[9] & Clemens R. Scherzer [1,2,4,11] ✉

Little is known about circular RNAs (circRNAs) in specific brain cells and human neuropsychiatric disease. Here, we systematically identify over 11,039 circRNAs expressed in vulnerable dopamine and pyramidal neurons laser-captured from 190 human brains and non-neuronal cells using ultra-deep, total RNA sequencing. 1526 and 3308 circRNAs are custom-tailored to the cell identity of dopamine and pyramidal neurons and enriched in synapse pathways. 29% of Parkinson's and 12% of Alzheimer's disease-associated genes produced validated circRNAs. *circDNAJC6*, which is transcribed from a juvenile-onset Parkinson's gene, is already dysregulated during prodromal, onset stages of common Parkinson's disease neuropathology. Globally, addiction-associated genes preferentially produce circRNAs in dopamine neurons, autism-associated genes in pyramidal neurons, and cancers in non-neuronal cells. This study shows that circular RNAs in the human brain are tailored to neuron identity and implicate circRNA-regulated synaptic specialization in neuropsychiatric diseases.

Initially cast aside as junk, circular RNAs are reemerging as an expanding, largely unexplored class of RNAs[1–3]. circRNAs are prominently enriched in synapses and brain, and linked to neuronal development[4,5] and aging[6,7]. Produced by the "back-splicing" of a downstream splice donor site to an upstream splice acceptor site, circRNAs have been linked to peripheral diseases, particularly cancer, but their role in brain health, is largely unexplored with initial clues pointing at a role in neurodegenerative diseases[8–10] and psychiatric diseases[11]. circRNA lack a polyadenylated tail and are considerably less prone to exonuclease-mediated degradation than mRNAs[12]. The stable circular transcripts have half-lives (18.8 to 23.7 h) that are at least 2.5 times longer than the median half-life of their linear counterparts (4.0 to 7.4 h)[13,14] and these properties may be important for their accumulation in terminally differentiated neurons.

High-performing neurons in the human brain are functionally and morphologically specialized to perform essential computation. Dopamine neurons in the midbrain control movements, mood, and

[1]APDA Center for Advanced Parkinson Disease Research, Harvard Medical School, Brigham & Women's Hospital, Boston, MA, USA. [2]Precision Neurology Program, Harvard Medical School and Brigham & Women's Hospital, Boston, MA, USA. [3]Genomics and Bioinformatics Hub, Harvard Medical School and Brigham & Women's Hospital, Boston, MA, USA. [4]Aligning Science Across Parkinson's (ASAP) Collaborative Research Network, Chevy Chase, MD 20815, USA. [5]State Key Lab of Digital Medical Engineering, School of Biological Science and Medical Engineering, Southeast University, Nanjing, China. [6]Department of Neurology, Zhejiang Hospital, Zhejiang, China. [7]School of Computer Science, Northwestern Polytechnical University, Xi'an, Shaanxi, China. [8]Department of Psychiatry, Brigham and Women's Hospital, Harvard Medical School, Boston, MA, USA. [9]Banner Sun Health Research Institute, Sun City, AZ, USA. [10]Department of Pathology, Brigham & Women's Hospital, Harvard Medical School, Boston, MA, USA. [11]Program in Neuroscience, Harvard Medical School, Boston, MA, USA. ✉e-mail: cscherzer@rics.bwh.harvard.edu

motivation, while pyramidal neurons in the temporal cortex play important roles in memory and language. Intriguingly, dopamine neurons are preferentially lost in Parkinson's disease[15], while pyramidal neurons are preferentially affected in Alzheimer's disease[16]. This raises a central question: How does the human genome program neuronal diversity? Could it be that brain-enriched circRNAs are tailored to neuron identity and contribute to their specialized synapses and disease vulnerability? Previous studies have examined circRNAs mostly in bulk tissues and lacked cellular resolution[5,8], and standard single-cell sequencing methods have been limited to linear and in many cases, polyadenylated RNAs. Intriguingly, one recent study suggested that circRNAs may display cell-specific expression patterns in inhibitory and excitatory neurons[17] by analyzing public full-length, single-cell RNAseq data comprising pilot human brain (e.g. glioblastoma) and organoid samples. Here we provide a comprehensive inventory of circRNAs in two prototypical types of brain cells—dopamine and pyramidal neurons— using laser-capture RNA sequencing that combines ultra-deep sequencing of polyadenylated and non-polyadenylated RNA with cell-type enrichment afforded by light microscopy[18]. We reveal that circRNA production is finely tuned to a cell's identity and prominent from Parkinson's, Alzheimer's disease, and other neuropsychiatric disease loci.

## Results

### Identification of 11,039 circRNAs with high confidence in human brain neurons

To systematically profile the transcriptome of human brain neurons, we have developed laser-capture RNA sequencing (lcRNAseq) to deeply sequence ribo-depleted total RNAs from laser-captured neurons for human post-mortem brains in the BRAINcode project[18]. Compared to most brain RNAseq studies, our method provides ultra-deep sequencing depth and reveals low-abundance transcripts as documented by our previous identification of transcribed non-coding elements (TNE)-defined putative enhancer RNAs (eRNAs)[18]. lcRNAseq captures total RNAs without limitation to polyA+ mRNAs, allowing us to detect non-polyadenylated RNAs such as circRNAs. Cell type enrichment is achieved through laser-capture microdissection. In this study, we doubled the sample size (n = 212) by including not just more healthy controls, but also brains from patients with Parkinson's disease (PD) and Alzheimer's disease (AD), the two most common neurodegenerative disorders (Supplementary Fig. 1a). After stringent quality control (Supplementary Fig. 1), 197 samples were available for downstream analysis (Fig. 1a, b). In brief, dopaminergic neurons (DA) from the midbrain substantia nigra pars compacta of 104 high-quality human brains (DA; healthy contols (HC): n = 59; prodromal PD-

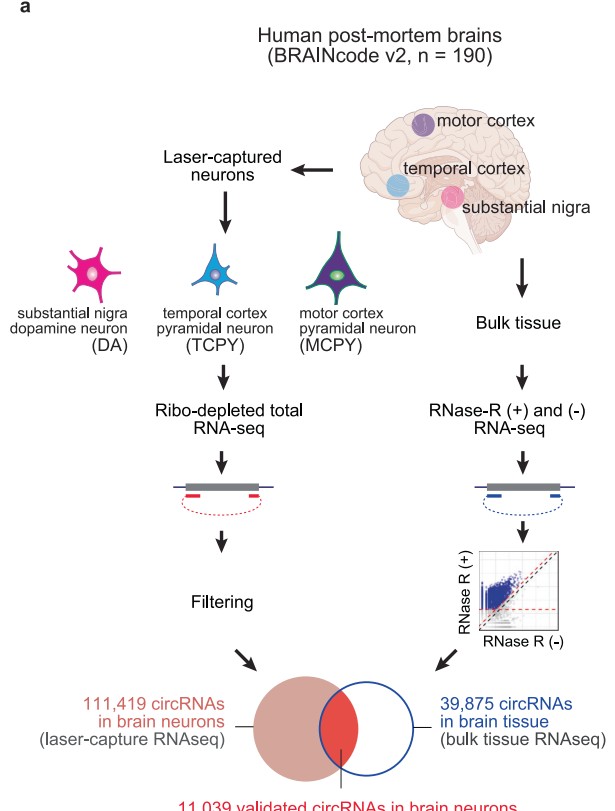

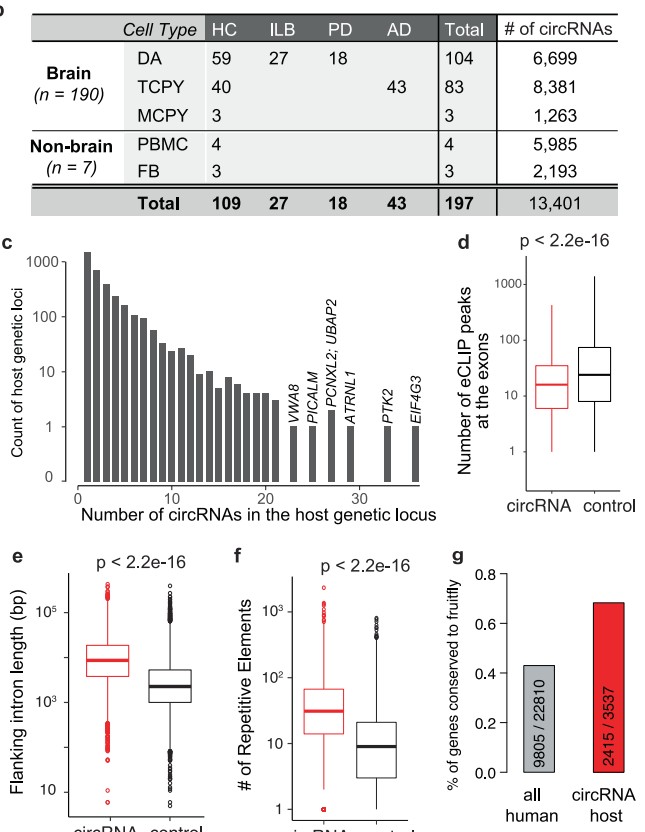

| Cell Type | | HC | ILB | PD | AD | Total | # of circRNAs |
|---|---|---|---|---|---|---|---|
| **Brain** (n = 190) | DA | 59 | 27 | 18 | | 104 | 6,699 |
| | TCPY | 40 | | | 43 | 83 | 8,381 |
| | MCPY | 3 | | | | 3 | 1,263 |
| **Non-brain** (n = 7) | PBMC | 4 | | | | 4 | 5,985 |
| | FB | 3 | | | | 3 | 2,193 |
| **Total** | | 109 | 27 | 18 | 43 | 197 | 13,401 |

**Fig. 1 | Identification of circular RNAs actively transcribed in human brain neurons. a** Schema of circRNA identification in laser-captured brain neurons. In total, for 190 human post-mortem brains we performed total RNAseq for three types of neurons that were laser-captured from different brain regions and computationally identified 111,419 circRNAs. Experimental validation with paired RNase R- treated and untreated bulk tissue RNAseq was used to confirm 11,039 enriched circRNAs as the final set of validated circRNAs used for downstream analyses in this study. The brain image in panel a is credited to Анна Богатырева / Adobe Stock - stock.adobe.com. **b** Overview of samples and expressed circRNAs in each cell type. **c** Number of circRNAs per host genetic locus. **d** There are significantly less

ENCODE-derived RNA-binding protein (RBP) eCLIP peaks overlapping with circRNAs (n = 10,845) vs. non-circularized exons (control, n = 10,845; two-sided Wilcox test, P < 2.2e−16). **e** Exons being circularized have longer flanking introns than background controls (n = 10,845; two-sided Wilcox test, P < 2.2e−16). **f** The flanking introns of circularized exons contain more repetitive elements than introns of control exons (n = 10,845; two-sided Wilcox test, P < 2.2e−16). **g** Proportions and numbers of human circRNA host genes conserved in *Drosophila melanogaster*. Grey bar shows the ratio of circRNA host genes to all genes in humans. Red bar shows the ratio of circRNA host genes to all genes in fruit fly.

associated Lewy body neuropathology (ILB): $n = 27$; PD: $n = 18$); pyramidal neurons from layers V/VI of the middle temporal gyrus of 83 brains (TCPY; HC: $n = 40$; AD: $n = 43$); and pyramidal neurons from the primary motor cortex of three brains (MCPY) were laser-captured (Fig. 1b). Human fibroblasts (FB) from four individuals and peripheral blood mononuclear white cells from three individuals (PBMC) were included as non-CNS cell types using the same pipeline. On average, we obtained 132 million RNAseq reads per sample, and intriguingly, ~20% of the RNAseq reads remained unmapped using a standard linear aligner (see Supplementary Data 1).

Recent development in bioinformatics algorithms allowed us to rescue many of the unmapped reads via mapping them chimerically in back-splicing order to discover novel circRNAs[19,20]. In this study, we first re-aligned the unmapped RNA-seq reads using Tophat-fusion[21] and then called circular RNAs using circExplorer (v2.0)[22] (see Methods). CircExplorer2 has been reviewed with the best overall performance balancing precision and sensitivity[19,20]. As in ref. 23, at least two unique back-spliced reads in overall samples were required for a circRNA to be considered for further evaluation. With this cutoff, we discovered an initial set of 111,419 circular RNAs expressed in neurons in overall 190 brain samples (Fig. 1a). circRNAs identified here were highly reproducible in replicate samples (Pearson's $r = 0.63$; Supplementary Fig. 1e).

## Independent validation of circRNAs with RNase R treated RNA-seq experiments

To experimentally confirm circular RNAs and to reduce false positives, we next used RNase R treatment of the human brain followed by RNAseq (see Method and Supplementary Data 2). RNase R is a 3′ to 5′ exoribonuclease that digests all linear RNAs (including the tail of intron lariats), but not circular RNA structures[12,24]. Paired RNase R-treated and -untreated RNAseq experiments were performed in six independent samples (see Method and Supplementary Fig. 2a). We rigorously required at least 20 unique reads per circular RNA in RNase R treated samples with at least two-fold enrichment compared to untreated samples in at least one pair (Supplementary Fig. 2a). Overall, out of the 111,419 circRNAs discovered in brain neurons, 11,039 met our stringent validation criteria; 48.0% of exon-derived circRNAs (e.g., 10,845 out of 22,593) and 0.22% of circular intronic RNAs (ciRNAs; 194 out of 88,826). These circRNAs were conservatively used for downstream analyses (Fig. 1a, validated circular RNAs in brain neurons). 98.2% of the validated circular RNAs were exon-derived circRNAs, suggesting that our filtering strategy effectively removed false-positive or lowly expressed circular RNAs (e.g., intron lariats from splicing[25]; Supplementary Fig. 2b, c). In total, we detected 6699, 8381, and 1263 circRNAs in dopamine neurons, pyramidal neurons, and Betz cells, respectively (Fig. 1b).

## Genomic characteristics of circRNAs in the brain

We observed that many genes transcribe multiple (up to 36) neuronal circRNAs (Fig. 1c). For example, the AD-associated gene *PICALM* produces 25 distinct circular RNA isoforms (Supplementary Fig. 2d). The number of circular RNAs per gene was linearly correlated with the number of exons of the host gene (Supplementary Fig. 2e). Circularized exons were significantly longer than the average exons in the human genome (Supplementary Fig. 2e), and had significantly less overlapping RNA binding protein (RBP) binding activity (using ENCODE eCLIP data[26]) than non-circularized exons (Fig. 1d). Similarly, the introns flanking circRNAs were longer; harbored significantly more repetitive elements; and had less overlapping RBP binding activity[26] compared to the introns flanking non-circularized exons (Fig. 1e, f, Supplementary Fig. 2e, see Methods). The host genes generating circRNAs are relatively more conserved (Fig. 1g). These might be universal features of circRNA biogenesis as they are highly consistent with prior evidence from non-neuronal cells[6,12,27–29].

## Dopamine and pyramidal neurons express cell type-specific circRNAs in human brain

To delineate circRNAs specifically expressed in each cell type, we further calculated the cell-specificity score of each circRNA based on the Jensen-Shannon distance of its expression profile similar as in ref. 23. CircRNAs with a specificity score $S \geq 0.5$ and mean expression > mean+s.d. of overall expression were defined as cell type-specific (see Methods). We focused our analysis of cell type-specificity on the control samples (i.e., 59 dopamine neuron samples, 43 pyramidal neuron samples, and 7 non-neuronal samples; excluding disease samples) to avoid confounding by disease state-driven changes. We identified 1526, 3308, and 4860 circRNAs preferential expressed in dopamine neurons, pyramidal neurons, and non-neuronal cells, respectively (left panel of Fig. 2a, Supplementary Data 3). Cell type-preferential expression of five dopamine neuron-specific, five pyramidal neuron-specific, and one non-neuronal cells-specific circRNA was confirmed in an independent set of substantia nigra and temporal cortex samples (e.g., the source regions for laser-captured dopamine and pyramidal neurons, respectively) using RNase R treatment and qPCR (Supplementary Fig. 3a). In total, 17 of 24 (70.8%) selected circRNAs showed consistent expression between preferential cell types and the source regions (Supplementary Data 4).

## Circular RNAs rather than the linear RNAs expressed from 3532 loci defined cell diversity and identity

3532 genetic loci custom-tailored 1526 cell type-specific circRNAs to the cell identity of dopamine neurons, 3308 circRNAs to pyramidal neurons, and 4860 cell type-specific circRNAs to non-neuronal cells, respectively (i.e., cell specificity score $S < 0.5$ and mean expression > mean + s.d.). Figure 2a, b; Supplementary Data 3). Surprisingly, the expression profiles of linear RNAs (e.g., mRNAs and long non-coding RNAs) transcribed from these same loci did not meet criteria for cell-type specificity (Fig. 2c) (i.e., $S \geq 0.5$ and mean expression > mean + s.d.) and cell-specificity scores of the linear transcripts were significantly lower than that of circRNAs (Fig. 2d). We observed the same trend even only considering the host gene with single circRNA (Supplementary Fig. 3c). Cell type-specificity of circRNAs was not explained by linear mRNA abundance from the corresponding loci. The abundance of total linear reads (i.e., spliced + unspliced reads) surrounding the back-splicing site only weakly correlated with the abundance of their associated circular reads (Spearman's rho = 0.10, see Supplementary Fig. 2f for detail). Interestingly, the circular-to-linear ratios of circRNAs were significantly higher in neurons vs. non-neuronal cells (Mann–Whitney test, $P < 2.2 \times 10^{-16}$, Fig. 2e). More research is needed to clarify whether this is modulated by distinct mechanisms regulating biogenesis, stability, and turnover of the circular RNAs, of their linear cognates, or both[27–29]. Interestingly, our data (Fig. 2a) suggest two types of cell type-specific circRNAs. First, some genetic host loci focus production of cell type-specific circRNAs onto a singular specific cell type (one locus producing one or more cell type-specific circRNAs specific to the same one cell type): These loci turn on circRNA production in one cell type and turn it off in the others e.g., 345 loci produce 400 circRNAs exclusively in dopamine neurons, 730 loci expressed 1046 circRNAs exclusively in pyramidal neurons, and 1191 loci expressed 2222 circRNAs exclusively in non-neuronal cells. Second, some loci precisely tailor the production of cell type-specific circRNAs for multiple cell types (e.g., via alternative back-splicing or combinations of different sets of exons) to the requirements of multiple types of cells (one locus producing multiple cell type-specific circRNAs for multiple cell types). Indeed, 306 super-host loci (Fig. 2a) tailored a diverse circRNAs portfolio specifically to dopamine neurons, pyramidal neurons, and non-neuronal cells via alternative back-splicing. These 306 super host loci expressed 478 distinct dopamine neuron-specific, 751 pyramidal neuron-specific, and 988 non-neuronal circRNA back-spliced variants (see Fig. 2a). RNA Binding Protein

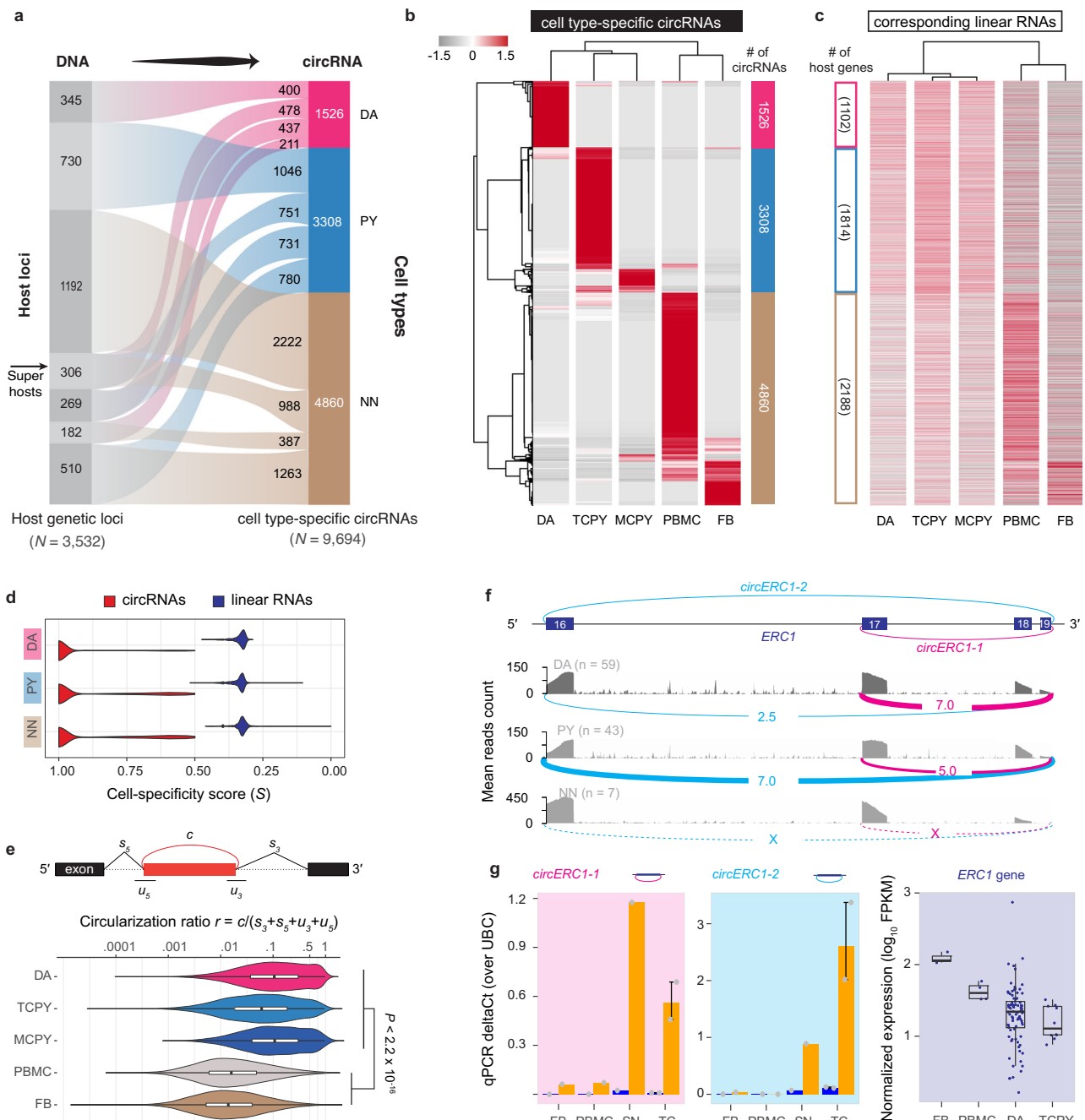

**Fig. 2 | Cell type-specificity of circular and linear transcripts diverges in human brain cells. a** The alluvial plot visualizes 3532 genetic loci that custom-tailored cell type-specific circRNAs to the cell identity of dopamine neurons (DA), pyramidal neurons (PY), and non-neuronal cells (NN), respectively (i.e., cell specificity score $S < 0.5$ and mean expression > mean + s.d.). **b**, **c** Expression heatmaps of cell type-specific circRNAs (b) and linear RNAs (c; e.g. mRNAs) from the corresponding loci in dopamine neurons (DA), pyramidal neurons of the temporal cortex (TCPY), pyramidal neurons of the motor cortex (MCPY), as well as peripheral blood mononuclear white blood cells (PBMC), and fibroblasts (FB). Cell type-specificity was defined based on cell type-specificity score and relative expression level (see Methods for details). Normalized expression values for circRNAs (in RPM) and for the linear RNAs (in FPKM) transcribed from the host loci were converted to z-scores for visualization purposes. CircRNAs and their corresponding linear mRNAs are shown in the same order in both panels. **d** Cell-specificity score distribution of circRNAs and linear mRNAs from the same genetic loci in three major cell types.

**e** Neuronal circRNAs showed a significantly higher circularization ratio (schema in upper panel) than non-neuronal circRNAs (two-sided Mann−Whitney test, $P < 2.2e$ −16). **f** circERC1-1 was preferentially expressed in dopamine neurons; circERC1-2 was preferentially expressed in pyramidal neurons; neither was expressed in non-neuronal cells. The genomic region visualized in the figure corresponds to chr12:1399017-1519619 (in hg19), which comprises exons 16 to 19 of the ERC1 gene transcript ENST00000542302. **g** Consistently, qPCR with RNase R indicated that circERC1-1 abundance was higher in substantia nigra (SN, the region containing dopamine neurons, $n = 1$ sample) compared to the temporal cortex (TC, the region containing pyramidal neurons, $n = 2$ biologically independent samples), where circERC1-2 was higher in the temporal cortex. Neither was meaningfully expressed in non-neuronal cells, e.g., fibroblasts (FB, $n = 1$) and white blood cells (PBMC, $n = 1$). ERC1 mRNA is actually expressed higher in the non-neuronal cells than in the neuronal cells. $N = 2$ technical replicates were analyzed per biological sample. Data are presented as mean values ± SEM.

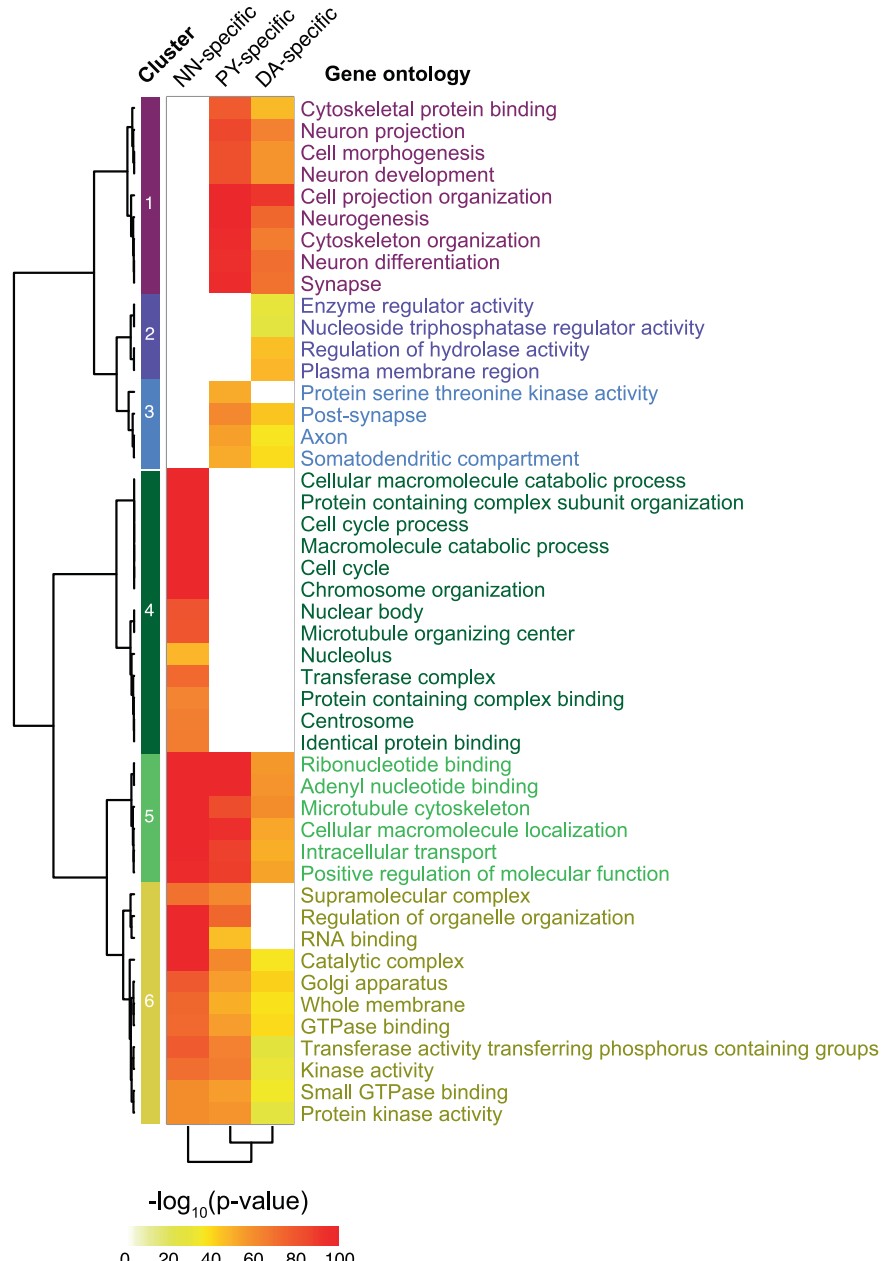

**Fig. 3 | circRNAs are predominantly expressed from synapse machinery.** Unbiased pathway analysis revealed that neuronal circRNA production was clustered around synapse function and neuronal projection loci, while in non-neuronal cells circRNA production clustered around cell cycle loci. Enriched Gene Ontology terms for the host genes of circRNAs specifically expressed in each cell type are shown (Fisher's exact test, Benjamini–Hochberg corrected $P < 0.05$).

activity superimposed from the ENCODE eCLIP track[26] highlighted subsets of RBPs statistically associated with these types of cell type-specific circRNAs (Supplementary Fig. 4). These super-host loci included the LKS/RAB6-interacting/CAST family member 1 (*ERC1*) locus. *ERC1* binds to the PD-linked RIM proteins[30,31], to facilitate the docking of synaptic vesicles at the presynaptic active zone. The *ERC1* locus expresses two distinct circRNAs, which we termed *circERC1-1* and *circERC1-2*, where *circERC1-1* is circularized between exon 17 and exon 19 and *circERC1-2* is alternatively back-spliced between exon 16 and exon 19 (see Fig. 2f). *circERC1-1* was preferentially expressed in dopamine neurons (Fig. 2f, g); *circERC1-2* was preferentially expressed in pyramidal neurons; neither was expressed in non-neuronal cells. Cell type-specific divergence of *circERC1-1* and *circERC1-2* was supported by qPCR with RNase R (Fig. 2g). *circERC1-1* abundance was higher in substantia nigra (the region containing dopamine neurons) compared

to the temporal cortex (the region containing pyramidal neurons). Neither was meaningfully expressed in non-neuronal cells, e.g., fibroblasts and white blood cells.

Thus, circRNA diversity provides finely tuned, cell type-specific information that is not explained by the corresponding linear RNAs from the same loci.

### circRNAs are predominantly expressed from synapse machinery (Fig. 3)

Unbiased pathway analysis of host loci revealed that cell type-specific neuronal circRNA production was clustered around synapse function and neuronal projection loci (Fig. 3), while in non-neuronal cells circRNA production clustered around cell cycle loci (Fig. 3). Thus, the host genes producing cell type-specific circRNAs in dopamine neurons vs. pyramidal neurons are actually representing similar synapse and

neuronal projection-related pathways. While it is reasonable to infer clues for candidate pathway membership from host gene enrichment analyses, careful experimental evaluation of individual circRNAs per se will be needed to substantiate their mechanistic roles in synapse specialization and disease. See Supplementary Data 5 for the full list of enriched functional terms.

## CircRNAs are linked to neuropsychiatric disease (Fig. 4)

We observed prominent expression of circRNAs from loci linked to neurodegenerative diseases (Fig. 4a). 80.2% of GWAS-derived AD candidate genes implicated by ref. 32 (174 of 217) produced one or more filtered circRNAs including *SORL1*[33], *MARK4*, *PICALM*, *PSEN1*, and *APP* (see Supplementary Fig. 5). 12.4% of AD GWAS genes (27 of 217) produced at least one rigorously validated circRNAs. Out of 109 genes implicated by the latest PD GWAS[30], 96 (88.1%) expressed one or more filtered circRNA transcripts and 32 (29.4%) expressed at least one validated circRNAs (Fig. 4a). Moreover, 99% of the AD-associated GWAS SNPs[32] (2334/2357) and 96% (6751/7057) of the PD-associated GWAS SNPs[30] were located in the proximity (i.e., within 1 Mbp) of a circRNA. These included pleomorphic genes linked to both Mendelian and sporadic forms of PD, e.g., *GBA, SNCA, RIMS1, RIMS2*[31], and *VPS13C* (see Supplementary Fig. 5). For example, under the peak of AD GWAS SNP rs867611 ($P = 2.19 \times 10^{-18}$), we detected 16 circRNAs actively expressed in the pyramidal neuron samples from the *PICALM* gene (Fig. 4b). Interestingly, the most abundantly expressed circRNA in both dopamine neurons and pyramidal neurons was *CDR1as* with over 70,000 total back-spliced reads in the neuronal samples. Previous studies showed that human *CDR1as* may serve as "sink" for miR-7 which represses the PD gene *SNCA*[34].

Unbiased enrichment analysis using the gene-disease annotations defined in DisGeNET[35] (updated for PD and AD GWAS-implicated susceptibility genes from refs. 30,32) showed that 20 neuropsychiatric diseases were statistically significantly enriched in circRNA loci compared to only three non-CNS diseases (Fig. 4c, Supplementary Fig. 2g, Supplementary Data 6) out of a total of 4638 human diseases and traits. Enrichment analysis showed a diverse spectrum of links between cellular circRNA loci and disease. PD- and intellectual disability-associated loci were enriched in circRNAs expressed in all three cell types (Fig. 4c). Examples of such loci include circRNAs from *KANSL1* gene (Fig. 4d), which was linked to PD by us[18] and others[36] as well as to neurodevelopmental disorders[37,38]. The enrichment of PD in circRNA-producing loci from all cell types is consistent with the increasingly appreciated roles of cortical neurons[31] and immune cells[39,40] in the pathobiology of PD, beyond classic dopaminergic neurodegeneration. Ten addiction traits related to drug and substance abuse were enriched in circRNAs active in dopamine neurons (see an example of circRNAs from addiction-related genes *PDE4B*[41,42] and *FTO*[41,43] in Fig. 4e and Supplementary Fig. 5). Autism and bipolar disease were enriched in circRNA loci actively expressed in pyramidal neurons. By contrast, the oncologic diseases leukemia and adenocarcinoma of the large intestine were enriched in circRNAs specific to non-neuronal cells (see an example of circRNAs from the cancer-associated gene *ATM*[44] in Fig. 4f). Surprisingly, atrial fibrillation, a common cardiac arrhythmia, was enriched in circRNA loci active in pyramidal neurons pointing to a potential role of synaptic plasticity in cardiac innervation[45].

We confirmed and evaluated circRNA expression from two of the implicated familial disease loci—*DNAJC6* and *PSEN1*—using a second method, qPCR with RNase R treatment. circ*PSEN1-2* is one of several circRNAs back-spliced from 4 exons of the familial AD gene *PSEN1* (Supplementary Figs. 3, 5). It was highly expressed in brain neurons with over 100 unique back-splice junction reads. qPCR confirmed that circ*PSEN1-2* is >20-fold enriched in both temporal cortex and substantia nigra after RNase R treatment (Supplementary Fig. 3).

Mutations in *DNAJC6* cause juvenile-onset, dopamine responsive, autosomal recessive PD[46–48]. *DNAJC6* plays a key role in uncoating the

clathrin-coated synaptic vesicles[49]. *circDNAJC6* produced from the locus was expressed highly in dopamine neurons, moderately in pyramidal neurons, but not in non-neuronal by lcRNAseq. Consistent with this, qPCR and RNase R treatment confirmed high expression in the substantia nigra, moderate expression in the temporal cortex, and minimal expression in the non-neuronal cells (Supplementary Fig. 3). Functionally, expression of *circDNAJC6* and several other synapse-related circRNAs (labeled in orange in Fig. 4g) was altered at the earliest, prodromal Parkinson's disease stages in brainstem-confined Lewy body Braak stages 1–3 ($P = 0.014$ by Wald test, with covariates of sex, age, PMI, and RIN adjusted, Fig. 4g, Supplementary Data 7). By contrast, expression of the linear mRNA *DNAJC6* was not affected (bottom right panel of Fig. 4g).

We also explored whether circRNAs expression changes during with disease progression. In PD-vulnerable dopamine neurons from 95 brains (with available neuropathology stages), 26 circRNAs (including *DNAJC6*) showed suggestive associations with progressive alpha-synuclein-positive Lewy body burden—spanning brainstem-predominant, midbrain, and cortical stages of PD (e.g., based on the Unified Lewy Body Staging System[50] of 0, I, IIa, III, and IV, see Supplementary Data 1). Furthermore, 51 circRNAs expressed in AD-vulnerable pyramidal neurons had suggestive associations with neurofibrillary tangle Braak stages with nominal $P$ values $\leq 0.05$ (Supplementary Fig. 6c).

## Discussion

Human brain cells custom tailor thousands of circRNAs to fit their cell identity. For many loci, which lacked diversity in linear mRNA expression, the corresponding "companion" circRNAs were cell-type specific. Dopamine and pyramidal neurons prominently expressed circRNAs from synaptic genes and non-neuronal cells prominently produced circRNAs from loci involved in cell cycle regulation.

The sheer number and diversity of circRNAs add a new class of components to the growing inventory of non-coding RNAs actively expressed in the human brain disease[18,51,52], including enhancer RNAs[18], microRNAs[33], and long-non-coding RNAs[51]. Indeed, there is reason to hypothesize that this expanding regulatory network of non-coding RNAs may be a major contributor to the exceptional diversity and performance of human brain cells that cannot be explained by the surprisingly small number of protein-coding genes in the human genome which are similar in humans and worms[52]. The fact that circRNAs are predominantly expressed from synapse loci in human dopamine and pyramidal neurons raises the possibility that they encode as yet unknow important functions in synaptic functions of the human neuronal networks controlling quintessential human experiences: fine motor movements, motivation, reward, and higher cortical functions. This expands on a postulated role for circRNAs in synaptosomes[4,5] and synaptic plasticity[4] in animals. What could that function be? It could be regulation of transcription consistent with the literature[2,4,25,53,54]. Alternatively, circRNAs—similar to some linear RNAs — might be targeted to the synapse as local regulatory switches that control the translation or assembly of highly specialized synaptic machinery for each type of neuron[55]. Their circular conformation may confer functional advantages (e.g., longer half-life) or suit transport along axons.

Importantly, 61% of all synaptic circRNAs were linked to brain disorders. Synaptic dysfunction—synaptopathy—may be one of the earliest defects in these neurodegenerative and neuropsychiatric diseases[56–61]. For example, in both toxic and genetic animal models of PD, synaptic plasticity is disrupted during the early phases of dopaminergic dysfunction, much earlier than nigral cell death and the clinical manifestation of motor features[57]. Disease-linked circRNAs expression showed, in part, evidence of cell type bias: Addiction-associated genes prominently express circRNAs in dopamine neurons, autism genes express circRNAs predominantly in pyramidal neurons, and

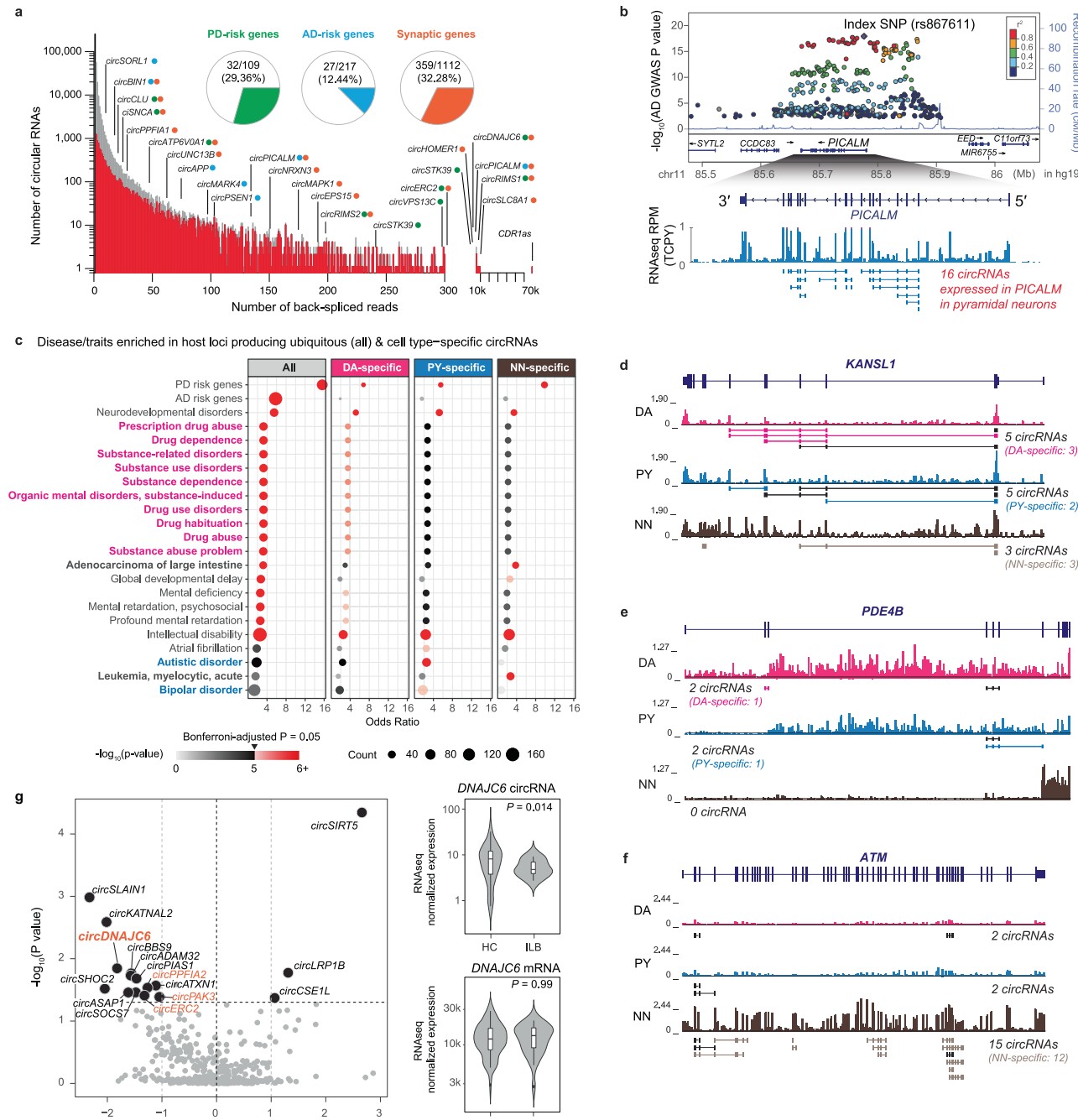

**Fig. 4 | CircRNAs are linked to neuropsychiatric disease. a** Histogram summarizing the number of *filtered* (grey bars) and *validated* (red bars) circRNAs (y-axis) supported by increasing numbers of back-spliced reads (x-axis). Pie charts show the proportion of PD-risk genes (green), AD-risk genes (cyan), and synaptic genes (orange) expressing *validated* circRNAs. Percentages for *filtered* circRNAs were 88.1% and 80.2% for AD- and PD-associated genes, respectively. Examples of circRNAs for each of these three categories (colored dots) are highlighted. **b** The AD GWAS locus *PICALM* produced 16 circRNAs in pyramidal neurons, 17 in dopamine neurons, and 12 in non-neuronal cells. GWAS *P* values on the *y*-axis of upper panel are derived from the AD GWAS study[32]. **c** Genes for addiction-associated traits (magenta font) preferentially produced circRNAs in dopamine neurons, autism- and bipolar-disease-associated genes (cyan font) preferentially in pyramidal neurons, and cancer loci (bold black font) preferentially in non-neuronal cells. Dot sizes correspond to the gene count in the query set, with color shade corresponding to the *P* values from Fisher's exact test. *P* values below the Bonferroni-corrected significance threshold of $1.08 \times 10^{-5}$ (e.g., 0.05 divided by the total number of 4368 unique disease sets tested) are considered statistically significant and highlighted

in red. **d**–**f** Examples of circRNAs expressed from the PD- and intellectual disability-associated gene *KANSL1* (d), from the addiction-associated gene *PDE4B* (e), and from the cancer-associated gene *ATM* (f). Normalized lcRNAseq density tracks in each cell type (DA: dopamine neuron; PY: pyramidal neurons; NN: non-neuronal cells) are shown below the gene model. circRNAs in each locus were identified using the healthy control samples (*n* = 109) **g** Functionally, expression of *circDNAJC6* and several other synapse-related circRNAs (labeled with orange dots) was reduced at the earliest, prodromal Parkinson's disease stages with brainstem-confined Lewy body neuropathology (*P* = 0.014, two-sided Wald test). By contrast, expression of the linear mRNA *DNAJC6* was not altered (*P* = 0.99, two-sided Wald test). Volcano plot indicates circRNAs differentially expressed in dopamine neurons from brains of individuals with prodromal PD-associated Lewy body neuropathology in the brainstem consistent with Lewy body Braak Stages 1–3 (ILB: *n* = 27) compared to controls (HC: *n* = 59). Circular RNAs were analyzed at the gene level, as indicated by their host gene symbol in the plot. The horizontal dashed line represents *P* = 0.05 and the vertical dashed lines (light grey) represent fold-changes of 0.5 and 2.

interestingly, PD GWAS-associated loci express circRNAs highly in non-neuronal cells as well as in neurons. Based on these and prior data[4,5,8,11,62], we hypothesize that circRNAs may serve as finely tuned, special-purpose RNA vehicles for the assembly of cell type-specific synapses and that their dysregulation may contribute to synaptopathies. The mechanisms controlling the biogenesis of cell type-specific circRNAs could involve subsets of RNA binding proteins (e.g. refs. [27–29] and Supplementary Fig. 4). Much more work will be required to fully elucidate the kinetics and relation of circular and cognate linear RNA biogenesis, the involved regulators, and to reveal how this complex RNA machinery specifies neuron identity and synapses.

More generally, this study provides a unique catalog of circRNAs in two major types of human brain neurons that will be generally useful for decoding genome function in neuropsychiatric disease and for advancing the burgeoning field of RNA medicines and diagnostics[63–65].

## Methods

### Sample collection and processing

In our BRAINcode project, we collected 190 high-quality, frozen postmortem human brain samples identified from Banner Sun Health Research Institute, Brain Tissue Center at Massachusetts General Hospital, Harvard Brain Tissue Resource Center at McLean Hospital, University of Kentucky ADC Tissue Bank, the University of Maryland Brain and Tissue Bank, Pacific Northwest Dementia and Aging Neuropathology Group (PANDA) at University of Washington Medicine Center, and Neurological Foundation of New Zealand Human Brain Bank. A subset of 96 of these samples were previously analyzed in ref. [18]. Detailed quality measures and demographic characteristics of these high-quality, frozen postmortem samples are shown in Supplementary Data 1. Median RNA integrity numbers (RIN) were 7.7, 7.4, and 7.3 for substantia nigra samples (used to laser-capture dopamine neurons), temporal cortex (used to laser-capture temporal cortex pyramidal neurons), and motor cortex samples (used to laser-capture Betz cells) indicating high RNA quality. Median post-mortem intervals were exceptionally short with 3 h for substantia nigra, 3 h for temporal cortex, and 13 h for motor cortex samples further consistent with the highest sample quality (Supplementary Data 1).

The 190 brain samples are composed of 102 samples from healthy control ("HC") subjects, 27 from incidental Lewy body cases ("ILB"), 18 from Parkinson's disease cases ("PD"), and 43 from Alzheimer's disease cases ("AD"). Healthy control subjects were defined with the following stringent inclusion and exclusion criteria. Inclusion criteria: (1) absence of clinical or neuropathological diagnosis of a neurodegenerative disease e.g., Parkinson's disease according to the UKPDBB criteria[66], Alzheimer's disease according to NIA-Reagan criteria[67], dementia with Lewy bodies by revised consensus criteria[68]. (2) PMI ≤ 48 h; (3) RIN[69] ≥ 6.0 by Agilent Bioanalyzer (good RNA integrity); (4) visible ribosomal peaks on the electropherogram. Exclusion criteria were: (1) a primary intracerebral event as the cause of death; (2) brain tumor (except incidental meningiomas); (3) systemic disorders likely to cause chronic brain damage. Incidental Lewy body cases are those not meeting clinical diagnostic criteria for PD or other neurodegenerative diseases but found with Lewy body inclusion at autopsy. ILB is widely considered a preclinical stage of PD[70], providing a unique opportunity to investigate preclinical molecular changes of PD. We also included seven non-brain tissue samples as controls, including four samples of peripheral blood mononuclear cell (PBMC) and three fibroblasts (FB), provided by Harvard Biomarker Study and Coriell Institute.

Our research complies with all relevant ethical regulations. This study was approved by the Institutional Review Board of Brigham and Women's Hospital.

### Laser-capture RNA-seq and RNase-treated RNA-seq

We applied the laser-capture RNA-seq (lcRNAseq) method to profile the total RNAs of brain neurons as we previously reported[18]. In brief, laser-capture microdissection was performed on the brain samples to extract neurons from different brain regions. For each substantia nigra sample, 300–800 dopamine neurons, readily visualized in HistoGene-stained frozen sections based on hallmark neuromelanin granules were laser-captured using the Arcturus Veritas Microdissection System (Applied Biosystems). For each temporal cortex (middle gyrus) or motor cortex sample, about 300 pyramidal neurons were outlined in layers V/VI by their characteristic size, shape, and location in HistoGene-stained frozen sections and laser-captured using the Arcturus Veritas Microdissection System (Applied Biosystems). Total RNA was isolated and treated with DNase (Qiagen) using the Arcturus Picopure method (Applied Biosystems), yielding approximately 7–8 ng RNA per subject. Total RNA was linearly amplified into 5–10 µg of double-stranded cDNA using the validated, precise, isothermal RNA amplification method implemented in the Ovation RNA-seq System V2 (NuGen)[71,72]. Unlike PCR-based methods that exponentially replicate original transcript and copies, with this method only the original transcripts are linearly replicated[71,72], and amplification is initiated at the 3' end as well as randomly thus allowing for amplification of both mRNA and non-polyadenylated transcripts[71,72]. Sequencing libraries were generated from 500 ng of the double-stranded (ds) cDNA using the TruSeq RNA Library Prep Kit v2 (Illumina) according to the manufacturer's protocol. The cDNA was fragmented, and end repair, A-tailing, adapter ligation was performed for library construction. Sequencing library quality and quantity control was performed using the Agilent DNA High Sensitivity Chip and qPCR quantification, respectively. Libraries were sequenced (50 or 75 cycles, paired-end) on an Illumina HiSeq 2000 and 2500 at the Harvard Partners Core.

To enrich the circRNAs, we also performed paired RNase R-treated and mock RNA-seq for 6 bulk tissues. To obtain sufficient amount of input RNAs, we mixed 1–6 samples from the same tissue type (see Supplementary Data 2). The total RNAs (10 µg) were treated with the Ribo-Zero Gold kit (Illumina) to remove rRNA before linear RNA digestion. The rRNA-depleted samples were purified using a modified RNeasy MinElute Cleanup kit (QIAGEN) method according to the Illumina's recommendation. After purification, the rRNA-depleted RNAs were divided into two equal parts: to one was added 2 µl 10× RNase R Reaction Buffer and 0.1 µl RNase R (20 U/µl, RNase R, Epicentre), the other received mock treatment with 2 µl 10× RNase R Reaction Buffer and 0.1 µl RNase-free water. We incubated the tubes at 37 °C for 30 min. After the digestion, the RNase R treated and mock treated RNAs were purified using RNeasy MinElute Cleanup kit (QIAGEN). Before the RNA-seq library preparation, about 10 ng of each the RNase digested and mock RNA samples were linearly amplified to yield enough amount of double-stranded cDNA with Ovation RNA-Seq System V2 kit (NuGen) according to the manufacture's instruction. After the linear amplification, -600 ng double-stranded cDNA samples were sonicated and applied to prepare the sequencing library using TruSeq RNA sample Preparation v2 kit (Illumina) following the manufacturer's instructions. Detailed characteristics of the samples used for RNase R experiment are shown in Supplementary Data 2.

### RNA sequencing data analysis pipeline

RNA-seq raw files in FASTQ format were processed in a customized pipeline. For each sample, we first filtered out reads that failed vendor check or are too short (<15nt) after removing the low-quality ends or possible adaptor contamination by using fastq-mcf with options of "-t 0 −x 10 −l 15 −w 4 −q 10 −u". We then checked the quality using FastQC and generated k-mer profile using kpal[73] for the remaining reads. Reads were then mapped to the human genome (GRCh37/hg19) using Tophat[74] (v2.0.8) by allowing up to 2 mismatches and 100 multiple hits. Reads mapped to ribosomal RNAs or to the mitochondrial genome were excluded from downstream analysis. Gene expression levels were quantified using FPKM (Fragments Per Kilobase of transcript per Million mapped reads). Only uniquely mapped reads were used to

estimate FPKM. To calculate normalized FPKM, we first ran Cuffquant[75] (v2.2.1) with default arguments for genes annotated in GENCODE (v19), and then ran Cuffnorm with parameters of "-total-hits-norm –library-norm-method quartile" on the CBX files generated from Cuffquant.

## Sample QC based on RNA-seq data

We performed sample QC similar to 't Hoen PA et al.[76]. In brief, we ran k-mer profiling for filtered reads using kpal[73] and calculated the median profile distance for each sample. Samples with distances clearly different from the rest samples were marked as outliers (Supplementary Fig. 1c). We also calculated pair-wise Spearman correlations of gene expression quantification across samples and measured the outlier via hierarchical clustering (Supplementary Fig. 1b). Moreover, we tested for concordance between reported clinical sex and sex indicated by the expression of the female-specific *XIST* gene and male-specific Y-chromosome gene *RPS4Y1* (Supplementary Fig. 1d). Samples from the first batch with a relatively low sequencing depth were also excluded. In addition to these samples used for cell type-specific transcriptome analyses, various additional control samples were analyzed (e.g., amplification controls, tissue homogenate), and technical replicates (Supplementary Fig. 1e). In the end, 197 out of 221 samples passed QC and are used for downstream analysis (Supplementary Fig. 1a).

## Calling circular RNAs

We first extracted the chimerically aligned RNAseq reads using Tophat-fusion[21] and then called circRNAs using circExplorer (v2.0)[22]. CircExplorer has been reviewed with the best overall performance in balance of precision and sensitivity[19,20]. To identify circRNAs that are not back-spliced from a canonical exon border, we ran circExplorer2 with a customized gene annotation file and the "--low-confidence" option. As in ref. 23, circRNAs with at least two unique back-spliced reads in overall samples are categorized as "being expressed" and kept as circRNA candidates. CircRNA expression is quantified by normalized reads per million (RPM) at the back-splicing sites for each sample.

## Cell specificity

The specificity score $S$ is defined as $S_{c,i} = 1 - \text{JSD}(p_c, \hat{q}_i)$, where JSD is the Jensen–Shannon distance, $p_c$ is the expression profile of a given circRNA $c$ expressed as a density of $(\text{RPM} + 1)$, and $\hat{q}_i$ is the unit vector of 'perfect expression' in a particular cell type $i$ (e.g. [1, 0, 0, 0, 0] for $i = 1$). Like the ref. 23, a circRNA is defined as cell-type specific if its specificity score $S \geq 0.5$ and mean expression in a cell type is larger than the mean + one standard deviation (s.d.) of the overall expression.

## circRNA-producing gene function enrichment analysis

Functional enrichment analysis of circRNA host genes was performed using the C5 gene sets (GO terms) implemented in the MSigDB database (version v7.2) using Fisher's exact test. Each gene set contains genes annotated to the same GO term. For each gene set, the Fisher's exact test was performed for $k$, $K$, $N-K$, $n$; where $k$ is the number of circRNA host genes that are part of a GO term gene set; $K$ is the total number of genes annotated to the same GO term gene set; $N$ is the total number of all annotated human genes in GENCODE (v19); and n is the number of genes in the query set. The top 10 GO terms in each GO category (BP, MF, and CC) enriched in these circRNA host genes are further slimmed using a similar algorithm as the clusterProfile package[77]. The slimmed results are shown in Supplementary Data 5 (all with an FDR $q$ value < 0.05).

We also evaluated whether there is a specific enrichment among circRNAs in genes associated with brain disorders. We used diseases in MeSH C10 (Nervous System Diseases) or F03 (Mental Disorders) for brain disorders, and associated diseases to genes using GenDisNet database. The disease-gene association was extracted from DisGeNet[35] (http://www.disgenet.org/) filtered with GDA score > 0.1. For all annotated protein-coding genes, we performed Fisher's exact test based on whether a gene is associated with a brain disorder and a gene hosts a circRNA (Supplementary Data 6).

## AD and PD risk genes

The GWAS-derived AD risk genes used in this study were extracted from Jansen et al.[32], where they defined AD potential causal genes with four gene-mapping strategies (nearest, eQTL, chromatin interaction, and GWGAS). Gene symbols were extracted from their Tables S13, S18 and there are 217 uniquely mapped genes. The GWAS-derived PD risk genes used in this study were extracted from Nalls et al.[30], where they defined PD potential causal genes based on nearest position and QTL. Gene symbols were extracted from their Table S2 and there are 109 uniquely mapped genes.

## Differentially expressed circRNAs

Similar to Dube et al.[8], we first aggregated all circular RNA reads count from one gene into a gene-based count matrix, and then used that count matrix as input to perform differential expression analysis using the DEseq2[78] framework. Covariates of sex, age, PMI (post-mortem interval), and RIN (RNA integrity number) were included in the negative binomial (NB) based generalized linear model, and $P$ values from the Wald test with covariates adjusted were reported.

## Confirming circRNA expression by qPCR

Quantitative PCR was performed using SYBR Green Master Mix (Thermo Fisher) on an ABI 7900HT instrument (Applied Biosystems). The divergent primer pairs flanking the back-splice sit were designed using Prime3 online primer design web tool (http://bioinfo.ut.ee/primer3-0.4.0/primer3/) and are shown in Supplementary Data 4. To confirm the expression of lcRNAseq-derived circRNAs in dopamine neurons and pyramidal neurons, relative abundances of target circRNAs were evaluated by qPCR in human substantia nigra or temporal cortex samples, as well as in human fibroblast and PBMC samples (shown in Supplementary Fig. 3). The human ubiquitin gene *UBC* was used as a reference to normalize RNA loading. Control samples lacking template and those lacking reverse transcriptase showed virtually no expression of these target circRNAs indicating that DNA contamination did not materially influence results. Expression values were analyzed using the comparative threshold cycle method[63]. All quantitative PCR reactions were conducted in triplicate. Equal amplification efficiencies for target and reference transcripts were confirmed using melting curve analysis.

## Data collection, statistical analysis, and data presentation

Sample sizes were based on the total number of available high-quality brain samples that met inclusion and exclusion criteria. No statistical methods were used to pre-determine sample sizes. No randomization of data collection was performed in this study. Brains were selected based on pre-defined inclusion and exclusion criteria (see above). Sample outliers were rationally identified as described in the Section on Sample QC based on RNA-seq data. Data were not excluded based on arbitrary post-hoc considerations. Data collection and analysis were not performed blinded to the conditions of the experiments.

R (The R Foundation for Statistical Computing, Vienna, Austria) was used for other statistical tests. Box plots were used to present multi-groups comparison. In all box plots, center line represents the median value; box limits, first and third quartiles; whiskers, the most extreme data point which is no more than 1.5 times the interquartile range from the box.

Statistical tests used in each figure: Fig. 1d–f, two-sided Wilcox test; Fig. 2e, two-sided Mann–Whitney test; Figs. 3, 4c, two-sided Fisher's exact test; Fig. 4g, two-sided Wald test; Supplementary Fig. 1e, f, Pearson's correlation test; Supplementary Fig. 2e, two-sided Wilcox test; Supplementary Fig. 2f, Spearman's rank correlation test; Supplementary Fig. 2g, two-sided Fisher's exact test; Supplementary Fig. 2h,

two-sided Student's *t* test; Supplementary Fig. 4b, two-sided Wilcox test; Supplementary Fig. 4c–e, two-sided Fisher's exact test; Supplementary Fig. 6b, d, Pearson's correlation test; Supplementary Data 5, 6, two-sided Fisher's exact test; Supplementary Data 7, two-sided Wald test.

## Reporting summary

Further information on research design is available in the Nature Portfolio Reporting Summary linked to this article.

## Data availability

The data supporting the findings of this study are available from the corresponding authors upon reasonable request. The RNA-seq raw FASTQ data and normalized expression matrix of all circRNAs in this study have been deposited in GEO under accession number GSE218203. The proceeded data for this study, including a browser track hub for all circRNAs from this study, can be queried at the BRAINcode project website at http://www.humanbraincode.org through a user-friendly interface.

## Code availability

Custom code associated with this study is publicly available at https://github.com/sterding/circRNA.

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

## Acknowledgements

This study was funded in part by NIH grant NIH R01AG057331, U01 NS082157, the U.S. Department of Defense (to C.R.S.), the American Parkinson Disease Association (APDA) Research Award (to X.D.). C.R.S.'s work is supported by NIH grants NINDS/NIA R01NS115144, U01NS095736, U01NS100603, and the American Parkinson Disease Association Center for Advanced Parkinson Research. X.D. received funding from the American Parkinson Disease Association (APDA). C.R.S and X.D.'s work was in part funded by the joint efforts of The Michael J. Fox Foundation for Parkinson's Research (MJFF) and the Aligning Science Across Parkinson's (ASAP) initiative. MJFF administers the grant [ASAP-000301] on behalf of ASAP and itself. This work was further enabled by NINDS U24 NS072026 National Brain and Tissue Resource for Parkinson's Disease and Related Disorders (to T.G.B.) and the National Institute on Aging (P30 AG19610 and P30AG072980, Arizona Alzheimer's Disease Center). We gratefully acknowledge the Banner Sun Health Research Institute, Massachusetts Alzheimer's Disease Research Center at Massachusetts General Hospital, Harvard Brain Tissue Resource Center at McLean Hospital, University of Kentucky ADC Tissue Bank, University of Maryland Brain and Tissue Bank, Pacific Northwest Dementia and Aging Neuropathology Group at University of Washington

Medicine Center, and Neurological Foundation of New Zealand for providing human brain tissue. For the purpose of open access, the author has applied a CC BY public copyright license to all Author Accepted Manuscripts arising from this submission.

## Author contributions

C.R.S. and X.D. conceived, designed, analyzed, and interpreted the study. X.D. performed bioinformatics analyses with contributions from R.B.M., T.W., A.E., and Y.B. Z.L., D.G. were responsible for laser-capture and lcRNAseq data production. Y.B., X.L., and Z.L. performed and analyzed RNase R and qPCR confirmation experiments. C.R.S. and X.D. wrote the manuscript with input from all other authors. C.R.S. supervised data production and analysis and contributed funding. T.B and G.S. neuropathologically characterized autopsy samples. M.B.F. contributed neuropathological expertise and critically reviewed results and manuscript. All authors reviewed, edited, and approved the manuscript prior to submission.

## Competing interests

C.R.S. has served as consultant, scientific collaborator or on scientific advisory boards for Sanofi, Berg Health, Pfizer, Biogen, and has received grants from NIH, U.S. Department of Defense, APDA, ASAP, and MJFF. X.D. has received funding from NIH, APDA, and ASAP. The other authors declare no competing interests.
