## [Peer Review File · Nature Communications]

Circular RNAs in the human brain are tailored to neuron identity and neuropsychiatric diseaseREVIEWER COMMENTS

Reviewer #1 (Remarks to the Author):

In the manuscript entitled “Circular RNAs in the human brain are tailored to cell identity”, the authors systematically identified the expression landscape of circular RNAs in vulnerable dopamine and pyramidal neurons using the lcrRNAseq method. The authors revealed the cell-type-specific expression of circRNAs in DA and PY neurons and further explored the relationship between circRNAs and neuropsychiatric diseases. In general, the manuscript provides a useful resource of neuron-specific circRNAs and could provide valuable insights into how circRNAs are associated with cell identity and affect neuropsychiatric diseases. However, some major concerns still need to be carefully addressed before accepted for publication.

Here are the specific comments:

(1) Page 3, lines 66-68. Although standard single-cell sequencing methods are often limited to linear and poly(A) RNAs, circRNAs can also be detected by some full-length scRNA-seq methods. A recent study also explored the cell-type expression of circRNAs (PMID: 35688820) which should be mentioned and discussed in context here.

(2) Page 4, lines 93-94. The authors stated that “~20% of the RNAseq reads remained unexplained using a standard linear aligner”. What’s the exact meaning of “unexplained”? The authors should give a precise definition (e.g., unmapped or chimeric aligned) here.

(3) What’s the percentage of back-splicing junction reads in each sample?

(4) Supplementary Fig. 2c. Interestingly, a large number of ciRNAs is detected in the dataset, most of which are lowly expressed and have lengths <1kb. What’s the percentage of intronic back-splicing events and intronic lariats in these circRNAs? Are these ciRNAs also specifically expressed between different neuron types?

(5) Page 6, lines 154-159. The authors found that the abundance of linear reads is weakly correlated with circular reads and concluded that “circRNAs are more stable than their cognate linear RNAs”. However, the weak correlation could be explained by the distinct regulation mechanism of the biogenesis of circular and linear RNAs, so I believe this statement should be carefully reconsidered.

(6) Page 6, lines 159-161. The two mechanisms described are more likely to be a description of results rather than real “mechanisms”. The authors could investigate deeper into the underlying regulating mechanism. For example, whether some splicing factors / RBPs are differentially expressed between different neurons and affect back-splicing of these cell type-specific circRNAs (e.g. PMID: 30893614).

(7) Fig5 d, e. Why are circRNAs derived from the same gene differentially expressed between DA/PY/NN

neurons? The authors could combine the eCLIP tracks from ENCODE to explore whether some RBPs have cell-type specific expression and possibly regulate the alternative back-splicing events.

(8) Page 7, line 190 & Page 8, line 211. Function enrichment analysis of host genes can only provide weak evidence of circRNA functions. Thus, the role of circRNAs in synaptic specialization and neuropsychiatric diseases should be carefully examined.

(9) In figure 1c-f, the authors summarized the characteristics of circRNAs detected in human brain neurons, including length distribution, flanking introns, repeats, etc. However, most of them have been reported elsewhere in previous studies. The authors may describe some novel characteristics of these neuron-specific circRNAs.

Reviewer #2 (Remarks to the Author):

Dong et al present their work on the characterization of circRNA populations in brains. In my opinion, it is methodologically sound and reads well. The data they have generated is very rich and meaningful to the community. Some small comments:

-The title is somewhat misleading, they do not isolate all cell types, but neurons. I would suggest changing to neurons.

-In the references cites to justify the use of circExplorer (16 & 17), there is no mention of circular RNAs or methodology. I would suggest performing some comparisons using DCC and findCirc. or remove the statement.

-Review the references, they seem to be shifted in some sections (see 16 & 17)

-In the statistical analyses, it might be adequate to adjust the results by sex and age, and probably post-mortem interval, since it is known to affect RNA quality. Given the complexity, unadjusted tests might led to false positive results.

-In the data availability, remember to add the GO number before publication, is currently missing

-When using the latest AD and PD GWAs, Jansen et al is not the latest AD GWAs. I would suggest using Bellenguez or Wightman.

-I feel like a missed opportunity to not check for AD/PD age at onset, Abeta, tau, alpha synuclein associations. Given the richness of the data generated, it will help support the claims about AD and PD preferential abundance.

Reviewer #3 (Remarks to the Author):

Dong et al. utilized the laser-capture RNA sequencing to systematically profile the transcriptome of human brain neurons, with a particular focus on circRNAs expressing in different types of neurons and non-neuronal cells. They concluded that cell type-specific circRNAs rather than the linear RNAs defines the diversity and identity of dopamine and pyramidal neurons and suggested that circRNAs are linked to neuropsychiatric diseases such as Parkinson's and Alzheimer's disease. Overall, it is an interesting study

and highlights the potential role of circRNAs in human brain neurons and the functional relevance to neuropsychiatric diseases. However, there are some conclusions which require further clarifications. Also, there are several important questions which should be addressed.

1.Lines 112-113: Out of the 111,419 circRNAs discovered in brain neurons, 11,039 were confirmed by RNase R. How about the other circRNAs? Undetectable or not validated? The validation rate seems quite low.

2.Lines 126-127: "The genes generating circRNAs are relatively more conserved." Is this observation due to the fact that the sequences of the flanking introns are more conserved?

3.Fig. 2a-c: As one or more circular RNAs can be produced from one host gene loci or cognate linear RNA, it is not fair to expect linear RNAs demonstrate the same cell type-specific pattern as that of circRNAs. Indeed, the authors suggested that the mechanism of "alternative back splicing-regulated circRNAs" contributes to the diversity of circRNAs among different cell types. If the authors only look at those host linear mRNAs which only produce one circular RNA, they should show the similar or same specificity as the circRNAs.

4.As RNA binding proteins can bind to the flanking introns of circRNAs to promote or block circRNA biogenesis, it is worthy for the authors to explore whether there are specific RBPs demonstrating differential expression or activity in specific types of neurons, contributing to the cell type-specific pattern of circRNAs in human brains.

5.Lines 157-159: the higher circular to linear ratio of circRNAs in neurons can also indicate a higher circular RNA biogenesis rate in neurons than non-neuronal cells. This possibility should not be ignored.

6.How about the "on-off regulated circRNAs" mechanism? What are the potential mechanisms controlling the on or off of host gene loci in terms of circRNA production? There was no further data or discussion about this mechanism.

7.The authors concluded that there was no much correlation between the abundance of total linear reads with the abundance of their derived circular reads, however, in Fig. 2f, non-neuronal cells demonstrated the higher expression of ERC1 mRNA and the lower expression of circERC1-1 and circERC1-2. This laid two questions:

1) whether the circularization of host linear mRNAs (resulting in circRNA production) leads to the decreased amount of mature linear mRNA transcripts?

2) whether the differential expression of linear mRNAs (as a result of the differential rate of circRNA production) rather than circRNAs defines the cell identity. Based on the current data, the authors cannot rule out the possibility that circRNAs are merely by-products of cell type-specific regulation of circularisation of host linear transcripts (e.g., transcribed from synapse pathway genes).

8.Fig. 4d: it could be simply due to the increased expression or activity of certain RBPs which bind to the flanking introns of different circKANSL1s and facilitate the circRNA production in DA and PY compared to NN.

Revised title:

Circular RNAs in the human brain are tailored to neuron identity and neuropsychiatric disease

Nature Communications Manuscript: NCOMMS-22-45680

Authors: Dong et al.

Point-by-point response to the Reviewers

Response to Reviewer 1:

In the manuscript entitled “Circular RNAs in the human brain are tailored to cell identity”, the authors systematically identified the expression landscape of circular RNAs in vulnerable dopamine and pyramidal neurons using the lcrRNAseq method. The authors revealed the cell-type-specific expression of circRNAs in DA and PY neurons and further explored the relationship between circRNAs and neuropsychiatric diseases. **In general, the manuscript provides a useful resource of neuron-specific circRNAs and could provide valuable insights into how circRNAs are associated with cell identity and affect neuropsychiatric diseases.** However, some major concerns still need to be carefully addressed before accepted for publication.

Concern 1. *Page 3, lines 66-68. Although standard single-cell sequencing methods are often limited to linear and poly(A) RNAs, circRNAs can also be detected by some full-length scRNA-seq methods. A recent study also explored the cell-type expression of circRNAs (PMID: 35688820) which should be mentioned and discussed in context here.*

RESPONSE: We appreciate the Reviewer’s suggestion and included a citation to this recent and intriguing work by Wu et al., 2022, based on the re-analysis of public SMARTer and SMART-seq2 data including brain samples (e.g. four glioblastoma samples, two epilepsy samples from the temporal lobe), and cultured organoids. We have updated the introduction accordingly.

We added this brief summary and citation to the revised manuscript: “Intriguingly, one recent study suggested that circRNAs may display cell-specific expression patterns in inhibitory and excitatory neurons¹⁷ by analyzing public full-length, single-cell RNAseq data comprising pilot human brain (e.g. glioblastoma) and organoid samples.”

Reference added: Wu et al., *Nature Communications*, 2022

Concern 2. *Page 4, lines 93-94. The authors stated that “~20% of the RNAseq reads remained unexplained using a standard linear aligner”. What’s the exact meaning of “unexplained”? The authors should give a precise definition (e.g., unmapped or chimeric aligned) here.*

RESPONSE: We clarified the sentence as follows: “On average, we obtained 132 million RNAseq reads per sample, and intriguingly, ~20% of the RNAseq reads remained *unmapped* using a standard linear aligner (see **Supplementary Table 1**).”

Concern 3. *What’s the percentage of back-splicing junction reads in each sample?*

RESPONSE: The percentage of back-splicing junction reads in each sample ranged from 1 to 5%. In the revised manuscript, the detailed values were added to the last column of **Supplementary Table S1**.

Concern 4. *Supplementary Fig. 2c. Interestingly, a large number of ciRNAs is detected in the dataset, most of which are lowly expressed and have lengths <1kb. What’s the percentage of intronic back-splicing events and intronic lariats in these circRNAs? Are these ciRNAs also specifically expressed between different neuron types?*

RESPONSE: According to CIRCEplorer2 algorithm, ciRNAs are defined by back-spliced reads that one end is at 5’ splicing site (5SS) and the other end is at the branch point (or in 5nt proximity if --no-fix option is set). Thus, all ciRNAs from CIRCEplorer2 are intronic lariats. Intronic back-splicing events which do not meet the criteria of ciRNAs (or intron lariats) are likely due to alignment error or incomplete gene annotation, and therefore not reported by CIRCEplorer2 and not included in this study.

ciRNAs were also specifically expressed between different neuron types. When we analyzed the cell specificity of circular RNAs, we included both intron lariat-derived ciRNAs and exon-derived circRNAs. 267, 41, and 31 ciRNAs were expressed in a cell type-specific manner in non-neuronal cells, pyramidal neurons, and dopamine neurons, respectively. The table below shows the breakdown of the 9,694 cell type-specific circRNAs shown in Fig. 2a into exon-derived circRNAs vs. ciRNAs. Moreover, the circType info was added to the **Supplementary Table S3**.

Cell type	Cell type-specific exon-derived circRNAs	Cell type-specific lariat-derived ciRNAs
NN	4593	267
PY	3267	41
DA	1495	31

Concern 5. Page 6, lines 154-159. The authors found that the abundance of linear reads is weakly correlated with circular reads and concluded that “circRNAs are more stable than their cognate linear RNAs”. However, the weak correlation could be explained by the distinct regulation mechanism of the biogenesis of circular and linear RNAs, so I believe this statement should be carefully reconsidered.

RESPONSE: We are grateful to the Reviewer for this perspicacious and thoughtful comment. We carefully reconsidered and revised this statement as follows:

“More research is needed to clarify whether this [observation] is modulated by distinct mechanisms regulating biogenesis, stability, and turnover of the circular RNAs, of their linear cognates, or both.”

Concern 6. Page 6, lines 159-161. The two mechanisms described are more likely to be a description of results rather than real “mechanisms”. The authors could investigate deeper into the underlying regulating mechanism. For example, whether some splicing factors / RBPs are differentially expressed between different neurons and affect back-splicing of these cell type-specific circRNAs (e.g. PMID: 30893614).

RESPONSE: First, we rephrased the wording in accordance with the Reviewer’s suggestion, e.g. using the descriptive wording of “two putative types of circRNAs” in the revised manuscript instead of “two putative mechanisms”.

Second, we more deeply investigated the potential underlying mechanism(s) generating these two types of cell type-specific circRNAs. RNA Binding Proteins have been implicated in the mechanisms regulating circRNA biogenesis. In accordance with the Reviewer’s excellent suggestion, we thus began to explore whether cell type-specific RBP expression could potentially account for cell type-specific circRNA expression. Fifteen RNA Binding Proteins (RBP) were expressed in a cell type-specific manner in non-neuronal cells, pyramidal neurons and dopamine neurons. However, no binding sites for these fifteen RBP were predicted in the 9,694 cell type-specific circRNAs here identified based on RBPmap. Thus, this initial exploration did not provide evidence for cell type-specific expression of RBPs as a driver of cell type-specific circRNA production.

Specifically, we extracted the superset of 3,470 known and identified RNA binding proteins (Gebauer

Cell Type	N	Cell-specific RBPs
NN	2	SLC25A6; HIST1H1B
PY	5	RBMY1F; CALML5; DDX53; RNASE13; RBMY1E
SNDA	8	RPL10L; RBMXL3; BOLA2; ZCCHC13; AKAP17A; RBMY1A1; RBMY1B; CALR3

et al. *Nature Reviews Genetics*, 2021) and determined their cell-specificity in our samples. We found that only 15 of the expressed RBP genes (15 out of 3,397) met our criteria for cell type-specific expression in our lcrRNAseq data set (see

Response Table below) consistent with previous reports indicating low tissue specificity for RBPs are tissue specific (Gerstbauer et al., *Nature Reviews Genetics*, 2014). However, none of these fifteen cell type-specific RBPs had predicted binding sites on the 9,694 cell type-specific circRNAs identified in our data based on RBPmap. We scanned the FASTA sequence of the 9,694 cell type-specific circRNAs in our dataset with the RBPmap software (Paz et al., *Nucleic Acids Research*, 2014). RBPmap includes a database of 274 RNA-binding motifs from human, mouse and *Drosophila melanogaster*. In total, 11,306,290 RBP binding sites were predicted on the cell-specific circRNAs with the default stringency (e.g., $P\text{-value}_{[\text{significant}]} < 0.01$ and $P\text{-value}_{[\text{suboptimal}]} < 0.02$). No predicted binding sites were identified on the 9,694 cell type-specific circRNAs for the fifteen RBPs expressed in a cell type-specific manner.

In addition to evaluating the role of cell type-specific expression of RBPs, we explored whether RBP binding activity as measured by eCLIP tracks in ENCODE might be associated with circRNA biogenesis. These new, exploratory analyses are detailed in the response to Concern 7 below.

Concern 7. Fig5 d, e. Why are circRNAs derived from the same gene differentially expressed between DA/PY/NN neurons? The authors could combine the eCLIP tracks from ENCODE to explore whether some RBPs have cell-type specific expression and possibly regulate the alternative back-splicing events.

RESPONSE: We are grateful for this interesting question and constructive suggestion. In accordance with the Reviewers suggestions, we computationally evaluated whether binding activity of RNA Binding Proteins (RBPs) might be linked to the biogenesis of various types of circular RNAs. Please note that we addressed the question on differential expression of RBPs in the Response to Concern 6 above.

First we evaluated the ENCODE eCLIP track (Van Nostrand, *Nature*, 2020), which represents the genome-wide binding peaks for a large collection of 139 RBPs in K562 and HepG2 cells. We superimposed the ENCODE RBP eCLIP peak positions onto the circRNAs identified in our study in human brain dopamine neurons, pyramidal neurons, and non-neuronal cells, on their flanking introns, and on control exons (i.e., random exons that do not form circRNAs; exactly matched numbers of exons; with exon length closely matched to our circRNAs). Interestingly, overall, circRNAs had less eCLIP peaks than non-circularized exons consistent with potentially generally reduced target RBP activity on the circularized RNAs. RBP binding activity on flanking introns was even lower. This new observation was included as a **new panel d in Fig. 1** of the revised manuscript. This panel is also shown adjacent here.

Supplementary Fig. 4. RNA Binding Protein (RBP) activity and cell type-specific circRNAs. a, Number of eCLIP peaks overlapping the two putative types of cell type specific circRNAs. b, Overlapped eCLIP binding peaks for 41 RBP were significantly positively (*NIP7*) or negatively enriched (all other RBPs) in circular exons of cell type-specific circRNAs for one cell type (e.g. circRNAs from locus producing *one or more cell type-specific* circRNAs specific to the *same one* cell type) compared to ubiquitous circRNAs with FDR < 0.05. Odds ratios from Fisher's exact test; $N = 5,608$ for circRNAs one cell type compared to $N = 1,942$ ubiquitous circRNAs. c, Overlapped eCLIP binding peaks for six RBPs were significantly less frequent at circular exons of cell type-specific circRNAs for multiple cell types (e.g., circRNAs from locus producing multiple cell type-specific circRNA for *multiple* cell types) compared to ubiquitous circRNAs. $N = 6,028$ circRNAs for cell type-specific circRNAs for multiple cell types compared to $N = 1,942$ ubiquitous circRNAs. d, Overlapped eCLIP binding peaks for 54 RBP were significantly positively ($N = 10$) or negatively enriched ($N = 44$) in circular exons of cell type-specific circRNAs for *one cell type* compared to cell type-specific circRNAs for *multiple cell types* with FDR < 0.05. Odds ratios from Fisher's exact test; $N = 5,608$ for circRNAs for *one cell type* compared to $N = 6,028$ circRNAs for *multiple cell types*.

Second, we then specifically overlapped the experimentally determined RBP activity from the ENCODE eCLIP track (Van Nostrand, *Nature*, 2020) with each of the two putative types of cell type-specific circRNAs observed: e.g. one locus producing one (or more) cell type-specific circRNAs specific to the same one cell type (**cell type-specific circRNAs for one cell type**) and one locus producing multiple cell type-specific

circRNA for *multiple* cell types (**cell type-specific circRNAs for multiple cell types**). Interestingly, the number of eCLIP peaks overlapping with circRNAs was significantly lower in the on/off type cell type-specific circRNAs than in those with alternative back-splice variants (**Supplementary Fig. 4a**). In this new analysis we found that eCLIP binding peaks for 41 RBP significantly overlapped with circular exons of cell type-specific **circRNAs for one cell type** compared to ubiquitous circRNAs with FDR < 0.05 (**Supplementary Fig. 4b**). eCLIP binding peaks for six RBPs were significantly less frequent at circular exons of **cell type-specific circRNAs for multiple cell types** compared to ubiquitous circRNAs (**Supplementary Fig. 4c**). Notably, eCLIP binding peaks for three RBP were reduced in both types of cell type-specific circRNAs: *IGF2BP1*, *IGF2BP2*, and *DDX51*. Binding peaks for 38 RBP were exclusively reduced in cell type-specific circRNAs for one cell type. Three RBP were exclusively reduced in cell type-specific circRNA for multiple cell types. Overlapped eCLIP binding peaks for 54 RBP were significantly positively ($N = 10$) or negatively enriched ($N = 44$) in circular exons of cell type-specific circRNAs *for one cell type* compared to cell type-specific circRNAs *for multiple cell types* with FDR < 0.05 (**Supplementary Fig. 4d**). As expected, eCLIP binding peaks for 39 of these 54 RBPs showed significant differences also in the above analyses using ubiquitous circRNAs as reference (**Supplementary Fig. 4b,c**). These 39 RBPs are underlined in panels b, c, and d.

Thus, these initial, exploratory data hint at a possible role for a subset of RBPs in tailoring the biogenesis of circRNAs for one cell type vs. circRNA for multiple cell types. Further research and direct experimental evidence – beyond the scope of our manuscript --- will be needed to evaluate these clues. *Overall, however, these data are consistent with the hypothesis that a subset of RBPs might be involved in tailoring circRNAs to neuron identity.*

Changes made:

1. We included the **new panel d** into the **main Fig. 1** of the revised manuscript.
2. We added the **new Supplementary Figure 4** to the Supplement.
3. We added pertinent **new References** implicating RBPs in circRNA biogenesis in the Discussion (e.g. Ji et al., *Cell Rep*, 2019; Okholm et al., *Genome Medicine*, 2020; Jiang et al., *Journal of Cancer*, 2021).
4. We revised the pertinent **Results and Discussion sections** in the manuscript. We also rephrased the wording in accordance with the Reviewers' suggestions, e.g. using the descriptive wording of "two putative types of circRNAs" in the revised manuscript instead of "two putative mechanisms".

Revised results section:

"Interestingly, our data (**Fig. 2a**) suggest two types of cell type-specific circRNAs. First, some genetic host loci focus production of *cell type-specific circRNAs onto a singular specific cell type* (*one locus producing one or more cell type-specific circRNAs specific to the same one cell type*): These loci turn on circRNA production in one cell type and turn it off in the others e.g., 345 loci produce 400 circRNAs exclusively in dopamine neurons, 730 loci expressed 1,046 circRNAs exclusively in pyramidal neurons, and 1,191 loci expressed 2,222 circRNAs exclusively in non-neuronal cells. Second, some loci precisely tailor the production of *cell type-specific circRNAs for multiple cell types* (e.g., via alternative back-splicing or combinations of different sets of exons) to the requirements of multiple types of cells (*one locus producing multiple cell type-specific circRNA for multiple cell types*). Indeed, 306 super-host loci (**Fig. 2a**) tailored a diverse circRNAs portfolio specifically to dopamine neurons, pyramidal neurons, and non-neuronal cells via alternative back-splicing. These 306 super host loci expressed 478 distinct dopamine neuron-specific, 751 pyramidal neuron-specific, and 988 non-neuronal circRNA back-spliced variants (see **Fig. 2a**). RNA Binding Protein activity superimposed from the ENCODE eCLIP track²⁶ highlighted subsets of RBPs statistically associated with these types of cell type-specific circRNAs (**Supplementary Fig. 4**)."

Revised Discussion section:

"The mechanisms controlling the biogenesis of cell type-specific circRNAs could involve subsets of RNA binding proteins (e.g. Refs. ²⁷⁻²⁹ and **Supplementary Fig. 4**). Much more work will be required to fully elucidate the kinetics and relation of circular and cognate linear RNA biogenesis, the involved regulators, and to reveal how this complex RNA machinery specifies neuron identity and synapses."

Concern 8. Page 7, line 190 & Page 8, line 211. *Function enrichment analysis of host genes can only provide weak evidence of circRNA functions. Thus, the role of circRNAs in synaptic specialization and neuropsychiatric diseases should be carefully examined.*

RESPONSE: We agree with the Reviewer's astute comment. Bioinformatic analyses of pathways enriched in the host genes of circRNAs is only an initial approximation as to potential functional roles of the implicated circRNAs. Careful experimental and mechanistic evaluation of the role of each circRNA of interest

in synapse specialization and in cellular and animal models of neuropsychiatric disease will be needed to draw more informed and more definitive conclusions on their functional roles. In accordance with the Reviewer's suggestion, we have included a sentence to clarify this important distinction and the limitation of the presented bioinformatics analysis to the readers.

The revised paragraph reads as follows:

“Unbiased pathway analysis of host loci revealed that cell type-specific neuronal circRNA production was clustered around synapse function and neuronal projection loci (**Fig. 3**), while in non-neuronal cells circRNA production clustered around cell cycle loci (**Fig. 3**). Thus, the host genes producing cell type-specific circRNAs in dopamine neurons vs. pyramidal neurons are actually representing similar synapse and neuronal projection-related pathways. *While it is reasonable to infer clues for candidate pathway membership from host gene enrichment analyses, careful experimental evaluation of individual circRNAs per se will be needed to substantiate their mechanistic roles, in synapse specialization and disease.*”

Concern 9. *In figure 1c-f, the authors summarized the characteristics of circRNAs detected in human brain neurons, including length distribution, flanking introns, repeats, etc. However, most of them have been reported elsewhere in previous studies. The authors may describe some novel characteristics of these neuron-specific circRNAs.*

RESPONSE: In accordance with the Reviewer's suggestion, we revised Fig. 1. We changed some of the panels describing circRNA characteristics. The new panels address the number of circRNAs per host gene locus (**Fig. 1c**); the number of eCLIP RBP activity peaks overlapping circular vs. non-circular RNAs in our data set **Fig. 1d**); and the proportion of circRNA host genes vs. non-host genes conserved in human and fruit fly (**Fig. 1g**). Moreover, as the Reviewer suggested, we moved the old panel Fig. 1d (indicating longer circRNA exon length as reported in previous studies) into the Supplement.

Response to Reviewer 2:

Dong et al present their work on the characterization of circRNA populations in brains. In my opinion, it is methodologically sound and reads well. The data they have generated is very rich and meaningful to the community.

Some small comments:

Concern 1. The title is somewhat misleading, they do not isolate all cell types, but neurons. I would suggest changing to neurons.

RESPONSE: We revised the title according to the Reviewer's suggestion. The revised title is "circular RNAs in the human brain are tailored to *neuron* identity and neuropsychiatric disease".

Concern 2. In the references cites to justify the use of circExplorer (16 & 17), there is no mention of circular RNAs or methodology. I would suggest performing some comparisons using DCC and findCirc or remove the statement. **Concern 3.** Review the references, they seem to be shifted in some sections (see 16 & 17).

RESPONSE: We thank the Reviewer for pointing out these two concerns that both pertain to shifted citations. We corrected and updated the citations.

Revised paragraph and revised citations read as follows:

"Recent development in bioinformatics algorithms allowed us to rescue many of the unmapped reads via mapping them chimerically in back-splicing order to discover novel circRNAs^{19,20}. In this study, we first re-aligned the unmapped RNA-seq reads using Tophat-fusion²¹ and then called circular RNAs using *circExplorer* (v2.0)²² (see **Methods**). CircExplorer2 has been reviewed with the best overall performance in balance of precision and sensitivity^{19,20}."

19. Hansen, T. B., Venø, M. T., Damgaard, C. K. & Kjems, J. Comparison of circular RNA prediction tools. *Nucleic acids research* **44**, e58 (2016).

20. Zeng, X., Lin, W., Guo, M. & Zou, Q. A comprehensive overview and evaluation of circular RNA detection tools. *PLOS Computational Biology* **13**, e1005420 (2017).

21. Kim, D. & Salzberg, S. L. TopHat-Fusion: An algorithm for discovery of novel fusion transcripts. *Genome Biology* **12**, 1–15 (2011).

22. Zhang, X. O. *et al.* Diverse alternative back-splicing and alternative splicing landscape of circular RNAs. *Genome Research* **26**, 1277–1287 (2016).

Concern 4. In the statistical analyses, it might be adequate to adjust the results by sex and age, and probably post-mortem interval, since it is known to affect RNA quality. Given the complexity, unadjusted tests might lead to false positive results.

RESPONSE: We agree with the Reviewer that adjusting for the covariates of sex, age, PMI, and RIN is particularly important for differential expression analyses in patient brain samples. We clarified the pertinent methods section. We adjusted the differential expression analyses for sex, age, PMI, and RIN. The reported p-values are based on covariates-adjusted analysis.

We clarified this methods paragraph:

"Differentially expressed circRNAs

Similar to Dube et al.⁸, we first aggregated all circular RNA reads count from one gene into a gene-based count matrix, and then used that count matrix as input to perform differential expression analysis using the DEseq2⁷⁵ framework. Covariates of sex, age, PMI (post-mortem interval), and RIN (RNA integrity number) were included in the negative binomial (NB) based generalized linear model, and P values from the Wald test with covariates adjusted were reported.

Concern 5. In the data availability, remember to add the GEO number before publication, is currently missing.

RESPONSE: The GEO accession number GSE218203 was added to the data availability section.

Concern 6. When using the latest AD and PD GWAS, Jansen et al is not the latest AD GWAS. I would suggest using Bellenguez or Wightman.

RESPONSE: GWAS as well as circRNAs are a very dynamic, fluid and evolving fields. Clearly, this will require multiple future iterations re-analyzing the relation of newly emerging GWAS loci and newly

emerging circRNAs. When we performed the current analysis, Jansen et al. Nature Genetics, 2019 was the latest, state-of-the-art AD GWAS and Bellenguez or Wightman GWAS were not available.

For our analysis, implicated AD susceptibility genes, not GWAS summary statistics are used to determine whether circRNAs are enriched in the GWAS-implicated host loci. Jansen and colleagues used four gene-mapping strategies (proximity, eQTL, chromatin interaction, and GWAS) to define a comprehensive, well substantiated set of putative AD genes that was confirmed (and extended by the Bellenguez or Wightman studies), well suited for our enrichment analysis. Of note, for PD, the GWAS used by Nalls et al., Lancet Neurology, 2019, is still the latest GWAS available.

In the revised manuscript, we thus appropriately rephrased and specified the pertinent wording to address the Reviewer's point. The revised sections more clearly specify, which AD and PD GWAS were here used. The clarified, more precise sentences read as follows:

“CircRNAs are linked to neuropsychiatric disease (Fig. 4). We observed prominent expression of circRNAs from loci linked to neurodegenerative diseases (**Fig. 4a**). 80.2% of GWAS-derived AD candidate genes **implicated by Ref. 29** (174 of 217) produced circRNAs including *SORL1*³⁰, *MARK4*, *PICALM*, *PSENI*, and *APP* (see Supplementary Fig. 4). Out of 109 genes implicated by the latest PD GWAS²⁷, 96 (88.1%) expressed one or more circRNA transcripts (**Fig. 4a**).” ...

“Unbiased enrichment analysis using the gene-disease annotations defined in DisGeNET³² (**updated for PD and AD GWAS-implicated susceptibility genes from Refs. 27,29**) showed that 20 neuropsychiatric diseases were statistically significantly enriched in circRNA loci compared to only three non-CNS diseases (**Fig. 4c, Supplementary Fig. 2g, Supplementary Table 6**) out of a total of 4,638 human diseases and traits.”

Concern 6. I feel like a missed opportunity to not check for AD/PD age at onset, Abeta, tau, alpha synuclein associations. Given the richness of the data generated, it will help support the claims about AD and PD preferential abundance.

RESPONSE: We are grateful to the Reviewer for this excellent suggestion. In the revised manuscript, accordingly, we explored the associations between circRNAs with alpha-synuclein-positive Lewy body neuropathology (Unified Lewy Body Staging System) and neurofibrillary tangle neuropathology (AD Braak stage), respectively.

Changes made: These results are summarized below and were **added** as new **Supplementary Results** and **Supplementary Fig. 6**. Moreover, a few sentences summarizing these exploratory analyses was added to the main manuscript.

Dopamine neuron circRNAs associated with Lewy body stages. We correlated the expression of circRNAs expressed in PD-vulnerable dopamine neurons with progressive Lewy body neuropathology using 95 (out of 104) dopamine neuron transcripomes with available neuropathology staging information (e.g., the Unified Lewy Body Staging System scores of 0, I, IIa, III, and IV, see Supplementary Table S1). 26 circRNAs had suggestive associations with Lewy body stage with nominal P values ≤ 0.05 using linear regression analysis adjusted for covariates of sex, age, RIN, and PMI (**Supplementary Fig. 6a**). We next compared the effect sizes of circRNAs associated with Lewy body stage to those circRNAs associated with the clinical diagnosis of PD and found that they were highly correlated in effect size and direction (Pearson's $r = 0.75$, $P \leq 2.22 \times 10^{-16}$; **Supplementary Fig. 6b**). In short, circRNAs associated with Lewy body stages were also associated with a clinical diagnosis of PD.

Pyramidal neuron circRNAs associated with AD neuropathology. We also correlated the abundance of circRNAs expressed in AD-vulnerable temporal cortex pyramidal neurons with the neuropathological AD Braak stages. We identified 51 circRNAs whose expression showed suggestive associations with neuropathological disease severity as indicated by nominal P values ≤ 0.05 (**Supplementary Fig. 6c**). Effect sizes of these circRNAs linked to AD neuropathology stages were also highly correlated with clinical diagnosis of AD (Pearson's $r = 0.91$, $P \leq 2.22 \times 10^{-16}$, **Supplementary Fig. 6d**).

Supplementary Figure 6. Exploring associations between circRNA expression and neuropathology. a, Dopamine neuron circRNAs associated with Lewy body stages. The abundance of 26 circRNAs was associated with Lewy body stage with nominal P values ≤ 0.05 using linear regression analysis adjusted for covariates of sex, age, RIN, and PMI; 95 (out of 104) dopamine neuron transcripts with available neuropathology staging information (e.g., the Unified Lewy Body Staging System scores of 0, I, IIa, III, and IV, see Supplementary Table S1). None achieved the multiple-testing-corrected significance threshold of $FDR \leq 0.05$. The line graph shows mean and standard error of the normalized expression counts of positively and negatively associated circRNAs. **b, Effect sizes of circRNAs associated with Lewy body neuropathology were highly correlated with the effect sizes for association with clinical diagnosis for the 18 samples with a clinical diagnosis of PD compared to 59 healthy controls without Lewy body neuropathology (Pearson's $r = 0.75$, $P \leq 2.22 \times 10^{-16}$).** Red dots, 32 suggestive circRNAs associated with Lewy body neuropathology or PD clinical diagnosis (e.g., $P < 0.05$ in either comparison); grey dots, circRNAs that were not associated with Lewy body neuropathology or PD clinical diagnosis or (e.g., $P \geq 0.05$ in both comparisons).

c, Pyramidal neuron circRNAs associated with AD neuropathology. The abundance of 51 circRNAs was associated with the neuropathological Braak stages of AD patients with nominal P values ≤ 0.05 using linear regression analysis adjusted for covariates of sex, age, RIN, and PMI ($N = 83$, including 9, 12, 11, 4, 4, 11, and 32 subjects with AD Braak stage of 0, 1, 2, 3, 4, 5, and 6, respectively; Fig. 1b). None achieved the multiple-testing-corrected significance threshold of $FDR < 0.05$. The line graph shows mean and standard error of

the normalized expression counts of positively and negatively associated circRNAs. **d, Effect sizes of circRNAs associated with neuropathological AD Braak stages were highly correlated with the effect sizes for association with clinical diagnosis of AD compared to healthy controls (Pearson's $r = 0.91$; $P \leq 2.22 \times 10^{-16}$; $N = 43$ AD and 40 control samples; Fig. 1b).** Red dots, 71 suggestive circRNAs associated with AD clinical diagnosis or AD neuropathology (e.g., $P < 0.05$ in either comparison); grey dots, circRNAs that are not associated with AD clinical diagnosis or AD neuropathology (e.g., $P \geq 0.05$ in both comparisons).

This was summarized in the revised manuscript as follows:

“We also explored whether circRNAs expression changes during with disease progression. In PD-vulnerable dopamine neurons from 95 brains (with available neuropathology stages), 26 circRNAs (including *DNAJC6*) showed suggestive associations with progressive alpha-synuclein-positive Lewy body burden --- spanning brainstem-predominant, midbrain, and cortical stages of PD (e.g., based on the Unified Lewy Body Staging System of 0, I, IIa, III, and IV, see **Supplementary Table S1**). Furthermore, 51 circRNAs expressed in AD-vulnerable pyramidal neurons had suggestive associations with neurofibrillary tangle Braak stages with nominal P values ≤ 0.05 (**Supplementary Fig. 6c**).”

Reviewer #3 (Remarks to the Author):

Dong et al. utilized the laser-capture RNA sequencing to systematically profile the transcriptome of human brain neurons, with a particular focus on circRNAs expressing in different types of neurons and non-neuronal cells. They concluded that cell type-specific circRNAs rather than the linear RNAs defines the diversity and identity of dopamine and pyramidal neurons and suggested that circRNAs are linked to neuropsychiatric diseases such as Parkinson's and Alzheimer's disease. Overall, it is an interesting study and highlights the potential role of circRNAs in human brain neurons and the functional relevance to neuropsychiatric diseases.

However, there are some conclusions which require further clarifications. Also, there are several important questions which should be addressed.

Concern 1. Lines 112-113: *Out of the 111,419 circRNAs discovered in brain neurons, 11,039 were confirmed by RNase R. How about the other circRNAs? Undetectable or not validated? The validation rate seems quite low.*

RESPONSE: We rigorously designed our experiment to prioritize identification of true positive circRNAs that can be replicated. To achieve this, we required conservative criteria for identification and confirmation of circRNAs. For examples, to reduce the number of false positive circRNAs we required that a circRNA must show substantial abundance (at least 20 back-splice junction reads in the RNase R experiments) AND at least 2-fold enrichment after RNase R-treatment vs. no RNase R treatment). See the red dash lines in the **Supplementary Figure 2a**. These two stringent criteria resulted in the confirmation of 48% of exon-derived circRNAs (e.g., 10,845 out of 22,593 circRNAs). By contrast, the confirmation rate for circular intronic RNAs, which are known to be produced with low abundance, was only 0.22% (ciRNAs; 194 out of 88,826). This is consistent with prior reports (e.g., Xiao et al., *Nucleic Acids Research*, 2019) indicating that much more circRNAs can be confirmed in replication experiments than ciRNAs (likely due to the lower abundance of ciRNAs). Moreover, because of our stringent confirmation criteria, low abundance circular transcripts, particularly, many ciRNAs were filtered out. Finally, it should be mentioned that some investigators uses less stringent validation criteria, e.g. only requiring that circRNAs have has equal or more reads in the RNase R compare to the mock group. With such relaxed criteria, 63.2% of exon-derived circRNAs (e.g. 14,057 out of 22,593 exon-derived circRNAs) would be considered confirmed in our data set. **Changes made:** We added a phrase to the revised manuscript clarifying this interesting point:

“Overall, out of the 111,419 circRNAs discovered in brain neurons, 11,039 met our stringent validation criteria; including 48.0% of exon-derived circRNAs (e.g. 10,845 out of 22,593) and 0.22% of circular intronic RNAs (ciRNAs; 194 out of 88,826).”

	Confirmed by RNase R	Not confirmed by RNase R	Total
circRNA	10845	11748	22593
ciRNA	194	88632	88826
total	11039	100380	111419

Concern 2. Lines 126-127: *“The genes generating circRNAs are relatively more conserved.” Is this observation due to the fact that the sequences of the flanking introns are more conserved?*

RESPONSE: *The evolutionary conservation observed appears chiefly driven by coding sequence.* Here we determined conservation of human circRNA-generating genes (i.e. “host genetic locus”) compared to their corresponding *Drosophila* orthologs using the Ensembl database (v74). Ensembl defines gene orthology for protein-coding genes based on protein trees of coding sequence alignment and comparisons. We did not specifically exam the conservation of the flanking introns. Interestingly, out of 3,537 human circRNA host genes, as many as 68% (2,415) had fly orthologs. By contrast, out of all 22,810 protein-coding genes in the human genome, only 42% (9,805) had a fly orthologs. A new panel was included in Fig. 1g to visualize this observation.

Concern 3. Fig. 2a-c: *As one or more circular RNAs can be produced from one host gene locus or cognate linear RNA, it is not fair to expect linear RNAs demonstrate the same cell type-specific pattern as that of circRNAs. Indeed, the authors suggested that the mechanism of “alternative back splicing-regulated circRNAs” contributes to the diversity of circRNAs among different cell types. If the authors only look at those*

host linear mRNAs which only produce one circular RNA, they should show the similar or same specificity as the circRNAs.

RESPONSE: We thank the Reviewer for this interesting idea. In accordance with the Reviewer's suggestion, we specifically examined linear RNAs from host loci which only produce one single circular RNA for cell specificity. We found that the cell-specificity scores of the linear RNAs from these host loci were low. circRNAs from these same loci had significantly higher cell-specificity scores. The plot was added as the new **Supplementary Fig. 3c** to the revised manuscript.

Concern 4. As RNA binding proteins can bind to the flanking introns of circRNAs to promote or block circRNA biogenesis, it is worthy for the authors to explore whether there are specific RBPs demonstrating differential expression or activity in specific types of neurons, contributing to the cell type-specific pattern of circRNAs in human brains.

RESPONSE: We thank Reviewer 3 for this intriguing comment. RNA Binding Proteins have been implicated in the mechanisms regulating circRNA biogenesis. In accordance with this Reviewer's excellent suggestion (and a similar comment by Reviewer 1), we thus began to explore *whether cell type-specific RBP expression could potentially account for cell type-specific circRNA expression*. Fifteen RNA Binding Proteins (RBP) were expressed in a cell type-specific manner in non-neuronal cells, pyramidal neurons and dopamine neurons. However, no binding sites for these fifteen RBP were predicted in the 9,694 cell type-specific circRNAs here identified based on RBPmap. Thus, this initial exploration did not provide evidence for *cell type-specific expression of RBPs as a driver of cell type-specific circRNA production*.

Specifically, we extracted the superset of 3,470 known and identified RNA binding proteins (Gebauer et al. *Nature Reviews Genetics*, 2021) and

Cell Type	N	Cell-specific RBPs
NN	2	SLC25A6; HIST1H1B
PY	5	RBMY1F; CALML5; DDX53; RNASE13; RBMY1E
SNDA	8	RPL10L; RBMXL3; BOLA2; ZCCHC13; AKAP17A; RBMY1A1; RBMY1B; CALR3

and determined their cell-specificity in our samples. We found that only 15 of the expressed RBP genes (15 out of 3,397) met our criteria for cell type-specific expression in our lcrRNAseq data set (see

Response Table below) consistent with previous reports indicating low tissue specificity for RBPs are tissue specific (Gerstbauer et al., *Nature Reviews Genetics*, 2014). However, *none of these fifteen cell type-specific RBPs had predicted binding sites on the 9,694 cell type-specific circRNAs identified in our data based on RBPmap*. We scanned the FASTA sequence of the 9,694 cell type-specific circRNAs in our dataset with the RBPmap software (Paz et al., *Nucleic Acids Research*, 2014). RBPmap includes a database of 274 RNA-binding motifs from human, mouse and *Drosophila melanogaster*. In total, 11,306,290 RBP binding sites were predicted on the cell-specific circRNAs with the default stringency (e.g., $P\text{-value}_{[\text{significant}]} < 0.01$ and $P\text{-value}_{[\text{suboptimal}]} < 0.02$). No predicted binding sites were identified on the 9,694 cell type-specific circRNAs for the fifteen RBPs expressed in a cell type-specific manner.

In addition to evaluating the role of cell type-specific *expression of RBPs*, we explored whether RBP *binding activity* as measured by eCLIP tracks in ENCODE might be associated with circRNA biogenesis. These new, exploratory analyses are detailed in the response to Concern 7 below.

Concern 5. Lines 157-159: the higher circular to linear ratio of circRNAs in neurons can also indicate a higher circular RNA biogenesis rate in neurons than non-neuronal cells. This possibility should not be ignored.

RESPONSE: We are grateful to the Reviewers 1 and 3 for pointing this out. We carefully reconsidered and revised this statement as follows:

"More research is needed to clarify whether this is modulated by distinct mechanisms regulating biogenesis, stability, and turnover of the circular RNAs, of their linear cognates, or both²⁷⁻²⁹."

Concern 6. How about the “on-off regulated circRNAs” mechanism? What are the potential mechanisms controlling the on or off of host gene loci in terms of circRNA production? There was no further data or discussion about this mechanism.

RESPONSE: First, we rephrased the wording in accordance with the Reviewer’s suggestion, e.g. using the descriptive wording of “two putative types of circRNAs” in the revised manuscript instead of “two putative mechanisms”.

Second, we more deeply investigated the potential underlying mechanism(s) generating these two types of cell type-specific circRNAs. RNA Binding Proteins have been implicated in the mechanisms regulating circRNA biogenesis. In accordance with the Reviewer’s excellent suggestion, we thus began to explore whether cell type-specific RBP expression could potentially account for cell type-specific circRNA expression. Fifteen RNA Binding Proteins (RBP) were expressed in a cell type-specific manner in non-neuronal cells, pyramidal neurons and dopamine neurons. However, no binding sites for these fifteen RBP were predicted in the 9,694 cell type-specific circRNAs here identified based on RBPmap. Thus, this initial exploration did not provide evidence for cell type-specific expression of RBPs as a driver of cell type-specific circRNA production.

Specifically, we extracted the superset of 3,470 known and identified RNA binding proteins (Gebauer

Cell Type	N	Cell-specific RBPs
NN	2	SLC25A6; HIST1H1B
PY	5	RBM11F; CALML5; DDX53; RNASE13; RBMY1E
SNDA	8	RPL10L; RBMXL3; BOLA2; ZCCHC13; AKAP17A; RBMY1A1; RBMY1B; CALR3

et al. *Nature Reviews Genetics*, 2021) and determined their cell-specificity in our samples. We found that only 15 of the expressed RBP genes (15 out of 3,397) met our criteria for cell type-specific expression in our lcrRNAseq data set (see

Response Table below) consistent with previous reports indicating low tissue specificity for RBPs are tissue specific (Gerstbauer et al., *Nature Reviews Genetics*, 2014). However, none of these fifteen cell type-specific RBPs had predicted binding sites on the 9,694 cell type-specific circRNAs identified in our data based on RBPmap. We scanned the FASTA sequence of the 9,694 cell type-specific circRNAs in our dataset with the RBPmap software (Paz et al., *Nucleic Acids Research*, 2014). RBPmap includes a database of 274 RNA-binding motifs from human, mouse and *Drosophila melanogaster*. In total, 11,306,290 RBP binding sites were predicted on the cell-specific circRNAs with the default stringency (e.g., $P\text{-value}_{\text{[significant]}} < 0.01$ and $P\text{-value}_{\text{[suboptimal]}} < 0.02$). No predicted binding sites were identified on the 9,694 cell type-specific circRNAs for the fifteen RBPs expressed in a cell type-specific manner.

In addition to evaluating the role of cell type-specific expression of RBPs, we explored whether RBP binding activity as measured by eCLIP tracks in ENCODE might be associated with circRNA biogenesis. These new, exploratory analyses are detailed in the response to Review #1’s Concern 7 below.

Reviewer #1’s Concern 7. Fig5 d, e. Why are circRNAs derived from the same gene differentially expressed between DA/PY/NN neurons? The authors could combine the eCLIP tracks from ENCODE to explore whether some RBPs have cell-type specific expression and possibly regulate the alternative back-splicing events.

RESPONSE: We are grateful for this interesting question and constructive suggestion. In accordance with the Reviewers suggestions, we computationally evaluated whether binding activity of RNA Binding Proteins (RBPs) might be linked to the biogenesis of various types of circular RNAs. Please note that we addressed the question on differential expression of RBPs in the Response to Concern 6 above.

First we evaluated the ENCODE eCLIP track (Van Nostrand, *Nature*, 2020), which represents the genome-wide binding peaks for a large collection of 139 RBPs in K562 and HepG2 cells. We superimposed the ENCODE RBP eCLIP peak positions onto the circRNAs identified in our study in human brain dopamine neurons, pyramidal neurons, and non-neuronal cells, on their flanking introns, and on control exons (i.e., random exons that do not form circRNAs; exactly matched numbers of exons; with exon length closely matched to our circRNAs). Interestingly, overall, circRNAs had less eCLIP peaks than non-circularized exons consistent with potentially generally reduced target RBP activity on the circularized RNAs. RBP binding activity on flanking introns was even lower. This new observation was included as a **new panel d in Fig. 1** of the revised manuscript. This panel is also shown adjacent here.

Supplementary Fig. 4. RNA Binding Protein (RBP) activity and cell type-specific circRNAs. **a**, Number of eCLIP peaks overlapping the two putative types of cell type specific circRNAs. **b**, Overlapped eCLIP binding peaks for 41 RBP were significantly positively (*NIP7*) or negatively enriched (all other RBPs) in circular exons of cell type-specific circRNAs for one cell type (e.g. circRNAs from locus producing *one or more cell type-specific* circRNAs specific to the *same one* cell type) compared to ubiquitous circRNAs with FDR < 0.05. Odds ratios from Fisher’s exact test; $N = 5,608$ for circRNAs one cell type compared to $N = 1,942$ ubiquitous circRNAs. **c**, Overlapped eCLIP binding peaks for six RBPs were significantly less frequent at circular exons of cell type-specific circRNAs for multiple cell types (e.g., circRNAs from locus producing multiple cell type-specific circRNA for *multiple* cell types) compared to ubiquitous circRNAs. $N = 6,028$ circRNAs for cell type-specific circRNAs for multiple cell types compared to $N = 1,942$ ubiquitous circRNAs. **d**, Overlapped eCLIP binding peaks for 54 RBP were significantly positively ($N = 10$) or negatively enriched ($N = 44$) in circular exons of cell type-specific circRNAs for *one cell type* compared to cell type-specific circRNAs for *multiple cell types* with FDR < 0.05. Odds ratios from Fisher’s exact test; $N = 5,608$ for circRNAs for *one cell type* compared to $N = 6,028$ circRNAs for *multiple cell types*.

Second, we then specifically overlapped the experimentally determined RBP activity from the ENCODE eCLIP track (Van Nostrand, *Nature*, 2020) with each of the two putative types of cell type-specific circRNAs observed: e.g. *one* locus producing *one (or more) cell type-specific* circRNAs specific to the *same one* cell type (**cell type-specific circRNAs for one cell type**) and *one* locus producing *multiple* cell type-specific circRNA for *multiple* cell types (**cell type-specific circRNAs for multiple cell types**). Interestingly, the number of eCLIP peaks overlapping with circRNAs was significantly lower in the on/off type cell type-specific circRNAs than in those with alternative back-splice variants (**Supplementary Fig. 4a**). In this new analysis we found that eCLIP binding peaks for 41 RBP significantly overlapped with circular exons of cell type-specific circRNAs for one cell type compared to ubiquitous circRNAs with FDR < 0.05 (**Supplementary Fig. 4b**). eCLIP binding peaks for six RBPs were significantly less frequent at circular exons of cell type-specific circRNAs for multiple cell types compared to ubiquitous circRNAs (**Supplementary Fig. 4c**). Notably, eCLIP binding peaks for three RBP were reduced in both types of cell type-specific circRNAs: *IGF2BP1*, *IGF2BP2*, and *DDX51*. Binding peaks for 38 RBP were exclusively positively reduced in cell type-specific circRNAs for one cell type. Three RBP were exclusively reduced in cell type-specific circRNA for multiple cell types. Overlapped eCLIP binding peaks for 54 RBP were significantly positively ($N = 10$) or negatively enriched ($N = 44$) in circular exons of cell type-specific circRNAs for *one cell type* compared to cell type-specific circRNAs for *multiple cell types* with FDR < 0.05 (**Supplementary Fig. 4d**). As expected, eCLIP binding peaks for 39 of these 54 RBPs showed significant differences also in the above analyses using ubiquitous circRNAs as reference (**Supplementary Fig. 4 b,c**). These 39 RBPs are underlined in panels b, c, and d.

Thus, these initial, exploratory data hint at a possible role for a subset of RBPs in tailoring the biogenesis of circRNAs for one cell type vs. circRNA for multiple cell types. Further research and direct experimental evidence – beyond the scope of our manuscript --- will be needed to evaluate these clues. Overall, however, these data are consistent with the hypothesis that a subset of RBPs might be involved in tailoring circRNAs to neuron identity.

Changes made:

1. We included the new panel d into the main Fig. 1 of the revised manuscript.
2. We added the new Supplementary Figure 4 to the Supplement.

3. We added pertinent **new References** implicating RBPs in circRNA biogenesis in the Discussion (e.g. Ji et al., *Cell Rep*, 2019; Okholm et al., *Genome Medicine*, 2020; Jiang et al., *Journal of Cancer*, 2021).

4. We revised the pertinent **Results and Discussion sections** in the manuscript. We also rephrased the wording in accordance with the Reviewers' suggestions, e.g. using the descriptive wording of "two putative types of circRNAs" in the revised manuscript instead of "two putative mechanisms".

Revised results section:

"Interestingly, our data (**Fig. 2a**) suggest two types of cell type-specific circRNAs. First, some genetic host loci focus production of *cell type-specific circRNAs onto a singular specific cell type (one locus producing one or more cell type-specific circRNAs specific to the same one cell type)*: These loci turn on circRNA production in one cell type and turn it off in the others e.g., 345 loci produce 400 circRNAs exclusively in dopamine neurons, 730 loci expressed 1,046 circRNAs exclusively in pyramidal neurons, and 1,191 loci expressed 2,222 circRNAs exclusively in non-neuronal cells. Second, some loci precisely tailor the production of *cell type-specific circRNAs for multiple cell types (e.g., via alternative back-splicing or combinations of different sets of exons) to the requirements of multiple types of cells (one locus producing multiple cell type-specific circRNA for multiple cell types)*. Indeed, 306 super-host loci (**Fig. 2a**) tailored a diverse circRNAs portfolio specifically to dopamine neurons, pyramidal neurons, and non-neuronal cells via alternative back-splicing. These 306 super host loci expressed 478 distinct dopamine neuron-specific, 751 pyramidal neuron-specific, and 988 non-neuronal circRNA back-spliced variants (see **Fig. 2a**). RNA Binding Protein activity superimposed from the ENCODE eCLIP track²⁶ highlighted subsets of RBPs statistically associated with these types of cell type-specific circRNAs (**Supplementary Fig. 4**)"

Revised Discussion section:

"The mechanisms controlling the biogenesis of cell type-specific circRNAs could involve subsets of RNA binding proteins (e.g. Refs. ²⁷⁻²⁹ and **Supplementary Fig. 4**). Much more work will be required to fully elucidate the kinetics and relation of circular and cognate linear RNA biogenesis, the involved regulators, and to reveal how this complex RNA machinery specifies neuron identity and synapses."

Concern 7. *The authors concluded that there was not much correlation between the abundance of total linear reads with the abundance of their derived circular reads, however, in Fig. 2f, non-neuronal cells demonstrated the higher expression of ERC1 mRNA and the lower expression of circERC1-1 and circERC1-2. This laid two questions: 1) whether the circularization of host linear mRNAs (resulting in circRNA production) leads to the decreased amount of mature linear mRNA transcripts? 2) whether the differential expression of linear mRNAs (as a result of the differential rate of circRNA production) rather than circRNAs defines the cell identity. Based on the current data, the authors cannot rule of the possibility that circRNAs are merely by-products of cell type-specific regulation of circularisation of host linear transcripts (e.g., transcribed from synapse pathway genes).*

RESPONSE:

We added a new paragraph at the end of the Discussion to address this:

"The mechanisms controlling the biogenesis of cell type-specific circRNAs could involve subsets of RNA binding proteins (e.g. Refs. ²⁷⁻²⁹ and **Supplementary Fig. 4**). Much more work will be required to fully elucidate the kinetics and relation of circular and cognate linear RNA biogenesis, the involved regulators, and to reveal how this complex RNA machinery specifies neuron identity and synapses."

Generally, our data show an association between circRNAs and cell type specificity (**Fig. 2b**). Moreover, our data show *no* general association between linear host mRNAs and cell type specificity (**Fig. 2c**). Thus, overall, the linear host mRNAs are unlikely to determine cell identity in our lcrNAseq data set. Moreover, overall, the circular-to-linear ratios of circRNAs were significantly higher in neurons vs. non-neuronal cells (Mann-Whitney test, $P < 2.2 \times 10^{-16}$, **Fig. 2e**), which is consistent with the specific ERC1 result mentioned.

To further evaluate the Reviewer's idea, we investigated the abundance of circular and linear RNAs from the ERC1 locus in the lcrNAseq data set. The left panel shows abundance of the two circRNAs (circERC1-1 and circERC1-2), while the right panel is for ERC1 mRNA itself. The circRNA ratio was higher in neurons vs. non-neuronal cells. The linear mRNA ratio was higher in non-neuronal cells vs. neurons. *However, we note that the ERC1 mRNA decrease in neurons vs non-*

neuronal cells is orders-of-magnitude smaller than the *ERC1* circRNA increase in neurons vs. non-neuronal cells. Taken together, our data show marked circRNA-linked cell type-specificity from these loci, but we cannot rule out the possibility that cell type-specific **regulation** of circularization of the cognate linear transcripts could contribute to the generation of cell type specific transcriptional programs.

Changes made: We added a new paragraph at the end of the Discussion to address this:

“The mechanisms controlling the biogenesis of cell type-specific circRNAs could involve subsets of RNA binding proteins (e.g. Refs. ²⁷⁻²⁹ and **Supplementary Fig. 4**). Much more work will be required to fully elucidate the kinetics and relation of circular and cognate linear RNA biogenesis, the involved regulators, and to reveal how this complex RNA machinery specifies neuron identity and synapses.”

Concern 8. Fig. 4d: it could be simply due to the increased expression or activity of certain RBPs which bind to the flanking introns of different *circKANS1s* and facilitate the circRNA production in DA and PY compared to NN.

RESPONSE: The objective of our manuscript is to delineate the universe of cell type-specific circRNAs in disease-relevant dopamine and pyramidal neurons from human brains. In Response to the Reviewers’ suggestions, we generally began to initially explore the potential association of RBPs with cell type-specific circRNA biogenesis as outlined in considerable detail in new analyses, figures, and results under Concerns 4 and 6. More detailed evaluations of the specific mechanisms causing the generation of each of the more than 9,694 cell type-specific circRNAs (and its relation to disease) will be important for future research. The resource here presented will provide the foundation for the field to perform such detailed, in depth, computational, experimental, and mechanistic characterization of specific, high-priority disease- and neurobiology-relevant circRNAs. This will require a major future research effort --- a gigantic task well beyond the scope of the current study.

Still, in response to the Reviewer’s comment we evaluated the predicted RBP binding sites at the flanking intron of *circKANS1s* using RBPmap (Van Nostrand, *Nature*, 2020) to scan the +/- 200bp flanking regions of *circKANS1s*. We found 10,076 predicted binding sites for 131 RBPs in the flanking regions (see the “*circKANS1.flanking200bp_RBPmap_Predictions*” track in the figure below). None of these 131 RBPs was expressed in a cell-type specific manner – consistent with the general lack of cell type-specificity observed for RBPs with predicted binding sites in the >9,000 cell type-specific circRNAs (see **Response to Concern 6** above). We further overlapped ENCODE eCLIP peaks (derived from K562 and HepG2 cell lines) with the flanking introns of *KANS1* circRNAs. 40 RBP eCLIP peaks overlapped with in the flanking introns of *KANS1* circRNAs (see the “*circKANS1.flanking200bp_ENCODE.eCLIP*” track in the figure below). We noted that some RBP peaks were specifically located at the cell type-specific *KANS1* circRNAs. For example, one QKI binding peak at K562 line is located at the flanking intron of a PY-specific circRNA. Several AQR binding peaks found at the flanking intron of a NN-specific circRNA. Again, these ENCODE eCLIP peaks were derived from K562 and HepG2 cell lines. How general this can be applied to the neuronal cells in our study is questionable and requires further investigation. Also note that none of the RBPs with binding sites in the *circKANS1* flanking introns were specifically expressed in our cell types (e.g. their cell-specificity score is below the cutoff of 0.5, see panel b in the figure below).

REVIEWER COMMENTS

Reviewer #1 (Remarks to the Author):

The revised manuscript is much improved, and all my concerns have been properly addressed. I'd like to recommend it to be published.

Reviewer #2 (Remarks to the Author):

The authors have made the requested changes. No more comments

Reviewer #3 (Remarks to the Author):

Overall, the authors have addressed the majority of concerns.

However, concern # 4 (similar to Reviewer 1, concern #7) needs further clarifications and/or analyses.

The authors concluded "However, no binding sites for these fifteen RBP were predicted in the 9,694 cell type-specific circRNAs here identified based on RBPmap." However, if I understand correctly, the authors used the circRNA sequences (i.e., circularized exonic sequences) but not their flanking intronic sequences for the analysis. Since it is well accepted that RBP can bind to the flanking introns of circRNAs to regulate backsplicing, the authors should look at the flanking introns instead.

To analyze the difference in RBP binding activity on the flanking introns between differentially expressed circRNAs identified in this study and unchanged circRNAs, the authors used control exons but not their flanking introns for the analysis (Fig. 1d). This is not a proper control.

Director, Precision Neurology Program
Brigham and Women's Hospital

Founder and Director,
Center for Advanced Parkinson Research
Harvard and Brigham and Women's Hospital

Clemens R. Scherzer, MD

Professor of Neurology
Harvard Medical School
Associate Neurologist

Brigham and Women's Hospital
Massachusetts General Hospital

Response to Reviewer #3 Comments on the Revised Manuscript

Title: Circular RNAs in the human brain are tailored to neuron identity and neuropsychiatric disease

Nature Communications Manuscript: NCOMMS-22-45680A

Authors: Dong et al.

REVIEWER COMMENTS

Reviewer #1 (Remarks to the Author):

The revised manuscript is much improved, and all my concerns have been properly addressed. I'd like to recommend it to be published.

Reviewer #2 (Remarks to the Author):

The authors have made the requested changes. No more comments

Reviewer #3 (Remarks to the Author):

Overall, the authors have addressed the majority of concerns. However, concern # 4 (similar to Reviewer 1, concern #7) needs further clarifications and/or analyses. The authors concluded "However, no binding sites for these fifteen RBP were predicted in the 9,694 cell type-specific circRNAs here identified based on RBPmap." However, if I understand correctly, the authors used the circRNA sequences (i.e., circularized exonic sequences) but not their flanking intronic sequences for the analysis. Since it is well accepted that RBP can bind to the flanking introns of circRNAs to regulate backsplicing, the authors should look at the flanking introns instead. To analyze the difference in RBP binding activity on the flanking introns between differentially expressed circRNAs identified in this study and unchanged circRNAs, the authors used control exons but not their flanking introns for the analysis (Fig. 1d). This is not a proper control.

Revised Response to Concern 4:

First, in accordance with the Reviewer's suggestion, we now evaluated **both flanking introns as well as circularized exons for putative RBP binding sites using RBPmap**. Fifteen RNA Binding Proteins (RBP) were expressed in a cell type-specific manner in non-neuronal cells, pyramidal neurons and dopamine neurons. As suggested by the Reviewer we scanned both the flanking introns of circRNAs as well as the circularized exons for putative binding sites. No binding sites for these fifteen RBP were predicted in the 9,694 cell type-specific circRNAs -- neither within the circularized exons nor the flanking introns.

Specifically, we extracted the superset of 3,470 known and identified RNA binding proteins (Gebauer et al. *Nature Reviews Genetics*, 2021) and determined their cell-specificity in our samples.

Cell Type	N	Cell-specific RBPs
NN	2	SLC25A6; HIST1H1B
PY	5	RBM1Y1F; CALML5; DDX53; RNASE13; RBMY1E
SNDA	8	RPL10L; RBMXL3; BOLA2; ZCCHC13; AKAP17A; RBMY1A1; RBMY1B; CALR3

We found that only 15 of the expressed RBP genes (15 out of 3,397) met our criteria for cell type-specific expression in our lcrNAseq data set (see Response Table above) consistent with previous reports indicating low tissue specificity for RBPs are tissue specific (Gerstbauer et al., *Nature Reviews Genetics*, 2014). However, *none of these fifteen cell type-specific RBPs had predicted binding sites on the flanking introns and circularized exons of the 9,694 cell type-specific circRNAs identified in our data based on RBPmap.* We scanned the FASTA sequence of the 9,694 cell type-specific circRNAs in our dataset, for both the circularized exons and the flanking introns, with the RBPmap software (Paz et al., *Nucleic Acids Research*, 2014). RBPmap includes a database of 274 RNA-binding motifs from human, mouse and *Drosophila melanogaster*. In total, 11,306,290 and 16,791,694 RBP binding sites were predicted on the cell-specific circRNA exons and flanking introns, respectively, with the default stringency (e.g., $P\text{-value}_{\text{significant}} < 0.01$ and $P\text{-value}_{\text{suboptimal}} < 0.02$). **No predicted binding sites in flanking introns and circularized exons were identified for the fifteen RBPs expressed for the 9,694 cell type-specific circRNAs.**

Second, we computationally estimated whether **binding activity** (ENCODE eCLIP tracks) **of RBPs on flanking introns and circularized exons** might be linked to the biogenesis of various types of circular RNAs.

We evaluated the ENCODE eCLIP track (Van Nostrand, *Nature*, 2020), which represents the genome-wide binding peaks for a large collection of 139 RBPs in K562 and HepG2 cells. As suggested by the Reviewer, we compared superimposed RBP eCLIP peaks on flanking introns of circRNAs to the peaks on flanking introns of matched control exons. We also similarly compared RBP eCLIP peaks on circularized exons vs. non-circularized control exons. Matched non-circularized “control” exons were random exons that do not form circRNAs; exactly matched for numbers of exons; with closely matched exon length; and their flanking introns.

Overall, flanking introns of circRNAs (Figure above) had less eCLIP peaks than the flanking introns of non-circularized “control” exons. Circularized exons of circRNAs also had less eCLIP peaks than the non-circularized “control” exons.

Third, we then more specifically overlapped the experimentally determined RBP activity from the ENCODE eCLIP track (Van Nostrand, *Nature*, 2020) with each of the two putative types of cell type-specific circRNAs observed: e.g. one locus producing one (or more) cell type-specific circRNAs specific to the same one cell type (**cell type-specific circRNAs for one cell type**) and one locus producing multiple cell type-specific circRNA for multiple cell types (**cell type-specific circRNAs for multiple cell types**).

There was no statistically significant difference in the number of eCLIP peaks overlapping with flanking introns of different types of circRNAs. However, the number of eCLIP peaks

overlapping with circularized exons was significantly lower in the class of circRNAs specific for one cell type than the class of **cell type-specific circRNAs for multiple cell types** (Wilcox's test $P = 1.1 \times 10^{-11}$).

Changes made:

We added a sentence summarizing the new analysis of eCLIP peaks in flanking introns outlined above to the main text. We added the new panel visualizing this result as revised **Supplementary Fig. 2e** (new right-most panel) and added panel **Supplementary Fig. 4a** to further illustrate eCLIP peaks in flanking introns and circularized exons. The revised sentence in the main text reads as follows:

"Similarly, the introns flanking circRNAs were longer; harbored significantly more repetitive elements; and had less overlapping RBP activity compared to the introns flanking non-circularized exons (**Fig. 1e-f, Supplementary Fig. 2e**, see **Methods**)."

REVIEWERS' COMMENTS

Reviewer #3 (Remarks to the Author):

The authors have addressed my concern. No further comments.